# Trehalose Phosphate Synthase Complex-Mediated Regulation of Trehalose 6-Phosphate Homeostasis Is Critical for Development and Pathogenesis in *Magnaporthe oryzae*

Xin Chen,[a,b] Yakubu Saddeeq Abubakar,[a,c] Chengdong Yang,[a] Xiaxia Wang,[a,d] Pengfei Miao,[a] Mei Lin,[a] Yuetong Wen,[a] Qiuqiu Wu,[a,b] Haoming Zhong,[a,b] Yuping Fan,[a,b] Meiru Zhang,[a] Zonghua Wang,[a,b,e] Jie Zhou,[a] ⬤Wenhui Zheng[a,b]

aState Key Laboratory of Ecological Pest Control for Fujian and Taiwan Crops, Fujian University Key Laboratory for Plant-Microbe Interaction, College of Plant Protection, Fujian Agriculture and Forestry University, Fuzhou, China
bKey Laboratory of Integrated Pest Management for Fujian-Taiwan Crops, Ministry of Agriculture, Fuzhou, China
cDepartment of Biochemistry, Ahmadu Bello University, Zaria, Nigeria
dFAFU-UCR Joint Center for Horticultural Biology and Metabolomics, Fujian Provincial Key Laboratory of Haixia Applied Plant Systems Biology, Fujian Agriculture and Forestry University, Fuzhou, China
eInstitute of Oceanography, Minjiang University, Fuzhou, China

Xin Chen and Yakubu Saddeeq Abubakar contributed equally to this article. Author order was determined by amount of leading contribution.

**ABSTRACT** Trehalose biosynthesis pathway is a potential target for antifungal drug development, and trehalose 6-phosphate (T6P) accumulation is widely known to have toxic effects on cells. However, how organisms maintain a safe T6P level and cope with its cytotoxicity effects when accumulated have not been reported. Herein, we unveil the mechanism by which the rice blast fungus *Magnaporthe oryzae* avoids T6P accumulation and the genetic and physiological adjustments it undergoes to self-adjust the metabolite level when it is unavoidably accumulated. We found that T6P accumulation leads to defects in fugal development and pathogenicity. The accumulated T6P impairs cell wall assembly by disrupting actin organization. The disorganization of actin impairs the distribution of chitin synthases, thereby disrupting cell wall polymer distribution. Additionally, accumulation of T6P compromise energy metabolism. *M. oryzae* was able to overcome the effects of T6P accumulation by self-mutation of its *MoTPS3* gene at two different mutation sites. We further show that mutation of *MoTPS3* suppresses MoTps1 activity to reduce the intracellular level of T6P and partially restore Δ*Motps2* defects. Overall, our results provide insights into the cytotoxicity effects of T6P accumulation and uncover a spontaneous mutation strategy to rebalance accumulated T6P in *M. oryzae*.

**IMPORTANCE** *M. oryzae*, the causative agent of the rice blast disease, threatens rice production worldwide. Our results revealed that T6P accumulation, caused by the disruption of *MoTPS2*, has toxic effects on fugal development and pathogenesis in *M. oryzae*. The accumulated T6P impairs the distribution of cell wall polymers via actin organization and therefore disrupts cell wall structure. *M. oryzae* uses a spontaneous mutation to restore T6P cytotoxicity. Seven spontaneous mutation sites were found, and a mutation in *MoTPS3* was further identified. The spontaneous mutation in *MoTPS3* can partially rescue Δ*Motps2* defects by suppressing MoTps1 activity to alleviate T6P cytotoxicity. This study provides clear evidence for better understanding of T6P cytotoxicity and how the fungus protects itself from T6P's toxic effects when it has accumulated to severely high levels.

**KEYWORDS** cell wall integrity, fungal development, *Magnaporthe oryzae*, trehalose 6-phosphate homeostasis, spontaneous mutation, TPS complex

Address correspondence to Jie Zhou, jiezhou@fafu.edu.cn, or Wenhui Zheng, wenhuiz@fafu.edu.cn.

Trehalose is a nonreducing sugar that consists of two glucose units that are jointed together by an $\alpha$-1,1-glycosidic bond. It is a structural component of most organisms, including plants, fungi, lichens, algae, bacteria, insects, and invertebrates, but not vertebrates (1, 2). Therefore, disruption of the trehalose biosynthesis pathway could be a promising alternative to developing safe and broad-spectrum fungicidal drugs, especially against human-pathogenic fungi (2, 3). To date, five different biosynthesis pathways have been established for the synthesis of trehalose, and the trehalose phosphate synthase/trehalose phosphate phosphatase (TPS/TPP) pathway is the most widely distributed trehalose biosynthesis pathway among these organisms. The TPS/TPP pathway consists of two key enzymes: trehalose 6-phosphate synthase (Tps1) and trehalose 6-phosphate phosphatase (Tps2). Tps1 catalyzes the reaction between UDPG-glucose and glucose-6-phosphate to form trehalose 6-phosphate (T6P), which is in turn dephosphorylated by Tps2 to produce a free trehalose molecule (4–6). In budding yeast, there are two regulatory subunits for the TPS/TPP pathway, named Tsl1p and Tps3p. Under stress, Tsl1p recruits Tps3p to form the TPS complex and thus induce the synthesis of trehalose. Under the stress recovery condition, trehalose biosynthesis is no longer required, and Tps3p is phosphorylated to inhibit the expression of Tps2p, leading to accumulation of T6P, which in turn inhibits Tps1p activity and trehalose biosynthesis (7–10).

Rice blast is one of the most devastating diseases of rice and accounts for 10 to 30% yield losses annually worldwide. The filamentous fungus *Magnaporthe oryzae* is the causative agent of the rice blast disease (11). Unlike in yeast, the TPS complex in *M. oryzae* only contains three subunits, including trehalose 6-phosphate synthase (MoTps1), trehalose 6-phospahte phosphatase (MoTps2), and a regulatory subunit (MoTps3) (12). A previous study indicated that a Δ*Motps1* mutant failed to synthesize trehalose and had defects in plant infection and growth on glucose as a carbon source (13). MoTps1 directly binds to NADPH to regulate a set of transcriptional corepressors, and the MoTps1-dependent genetic switch is essential for the initiation of rice blast disease (14). The glucose-6-phosphate sensor MoTps1 was later shown to trigger the carbon catabolite repression pathway via inactivation of Nmr1-3 (15). In addition, MoTps3 is important for the activation of MoTps1, and like in the Δ*Motps1* mutant, deletion of *MoTPS3* also leads to a defect in plant infection (12).

Unlike MoTps1 and MoTps3, the roles of MoTps2 and its relationships with other TPS complex subunits have not been studied in *M. oryzae*. Nevertheless, *TPS2* orthologs are well conserved in other fungi, and several of them have been characterized in fungal pathogens. In *Fusarium graminearum*, loss of the *FgTPS2* gene significantly affects the growth, sporulation, and virulence of the fungus, and the Δ*Fgtps2* mutant is defective in mycotoxin production (16). In *Ustilago maydis*, a basidiomycete plant pathogen, UmTps2 is important for plant infection and responses to osmotic stress and high temperatures for trehalose accumulation (17, 18). In the human-pathogenic fungi *Candida albicans* and *Cryptococcus gattii*, Tps2 is important for stress response and plays an important role in virulence (19–21). In *Aspergillus fumigatus*, the *TPS2* ortholog named *OrlA* plays an important role in regulating glycolytic flux, cell wall integrity, and virulence (22). It has been widely believed that loss of *TPS2* homologue leads to accumulation of T6P, which is toxic to the cell. We therefore believe that Tps2 should be very important in *M. oryzae*, and that understanding the roles and mechanisms of MoTps2 functions and how T6P homeostasis is regulated would provide a better way to develop effective fungicides to control the devastating rice blast disease.

In the present study, we used genetic approaches to analyze the biological functions of MoTps2 in the rice blast fungus. Our results indicate that deletion of *MoTPS2* leads to severe defects in growth, conidiation, and pathogenicity. MoTps2 was shown to regulate chitin deposition on cell walls via the cytoskeleton and localization of chitin synthases. In addition, deletion of *MoTPS2* results in T6P accumulation, which negatively impacts the physiologic balance of some glycolysis and tricarboxylic acid (TCA) cycle metabolites, as well as amino acid biosynthesis. In addition, the TPS complex

possesses a spontaneous "correction" way to modulate T6P homeostasis by spontaneous mutation of the regulatory subunit MoTps3. This spontaneous correction function further suppresses the MoTps1 activity to downregulate T6P, resulting in partial restoration of the Δ*Motps2* mutant's defects in growth, conidiation, and pathogenicity. This study presents new insights into the cytotoxicity of T6P accumulation and uncovers a spontaneous way of correcting of T6P accumulation cytotoxicity in *M. oryzae*.

## RESULTS

**MoTps2 is critical for vegetative growth, conidiogenesis, and pathogenicity in *M. oryzae*.** *MoTPS2* (MGG_03441) is the only *TPS2* homologue in *M. oryzae*, and it encodes a 1,020-amino-acid (aa) protein that shares 33.4% identity in amino acid sequences with *Saccharomyces cerevisiae* Tps2p (23). Phylogenetic analysis and domain alignment suggest that *TPS2* homologues are conserved in fungi (see Fig. S1 in the supplemental material). To investigate its biological role in *M. oryzae*, we generated *MoTPS2* deletion mutants (henceforth denoted as Δ*Motps2*) by replacing *MoTPS2* with a hygromycin resistance cassette (*HPH*), using homologous recombination, and further confirmed the deletion by PCR analysis and Southern blot assay (see Fig. S2a in the supplemental material). Furthermore, we generated the complemented (-com) strain by expressing the MoTps2-GFP (green fluorescent protein) fusion protein in the Δ*Motps2* mutant background. The wild type strain (70-15), mutant (Δ*Motps2*), and complemented strain (Δ*Motps2*-com) were then cultured on complete medium (CM), minimal medium (MM), and rice bran medium (RBM) for 7 days to compare their growth rates. The Δ*Motps2* mutant showed significant reductions ($P < 0.01$) in growth rate on these media compared to the 70-15 and Δ*Motps2*-com strains (Fig. 1a and Table 1). Because the Δ*tps1* mutant has a growth defect on glucose and other fermentable sugars (2), we also assayed for the growth of the Δ*Motps2* mutant on MM and found that its severe growth defect is not altered by different carbon sources (see Fig. S3 in the supplemental material).

In comparison with the wild-type and complemented strains, the Δ*Motps2* mutant had a significant ($P < 0.01$) reduction in conidiation (Table 1). Microscopic observations showed that it was defective in conidiophore development and conidium formation (Fig. 1b). The Δ*Motps2* mutant also displayed abnormal conidial morphology; most of its conidia possess only one or no septum (Fig. 1b). We then went further to test the pathogenicity of the deletion mutant. Due to the poor conidium production of the Δ*Motps2* mutant, hyphal blocks were used for inoculation on intact and wounded barley leaves to assay for the pathogenicity. The Δ*Motps2* mutant failed to produce lesions on intact barley leaves (Fig. 1c). Close microscopic examination showed that the Δ*Motps2* mutant failed to produce invasive hyphae inside host cells, even at 36 h postinoculation (Fig. 1d). However, some small lesions were observed when mycelial blocks from the mutant were inoculated on wounded barley leaves (Fig. 1c), suggesting that MoTps2 is required for normal pathogenicity and host cuticle penetration. The subcellular localization of MoTps2 was also investigated by expressing MoTps2-GFP fusion protein in the Δ*Motps2* mutant. Confocal microscopic examination indicated that MoTps2 localizes to cytoplasm at different developmental stages of the fungus (Fig. 1e). Taken together, our results suggest that *MoTPS2* encodes a cytoplasmic protein essential for vegetative growth, conidiogenesis, and pathogenicity in *M. oryzae*.

**MoTps2 is important for development of infection structures and generation of turgor pressure.** To successfully breach the host cuticle and gain entry into the host cell, *M. oryzae* forms a dome-shaped structure called the appressorium at the end of its germ tube; glycerol is accumulated in such a structure to generate turgor pressure used to physically rupture the cuticle (24). To explore the role of MoTps2 in appressorial development, conidia from the wild type and Δ*Motps2* mutant were inoculated on artificial hydrophobic surfaces for germ tube formation and subsequent appressorial development. Over 92% of the wild-type conidia developed germ tubes and appressoria at 4, 6, and 24 h postincubation (hpi). The Δ*Motps2* mutant was normal in germination, but only about 2.5% of its germ tubes formed appressoria at 24

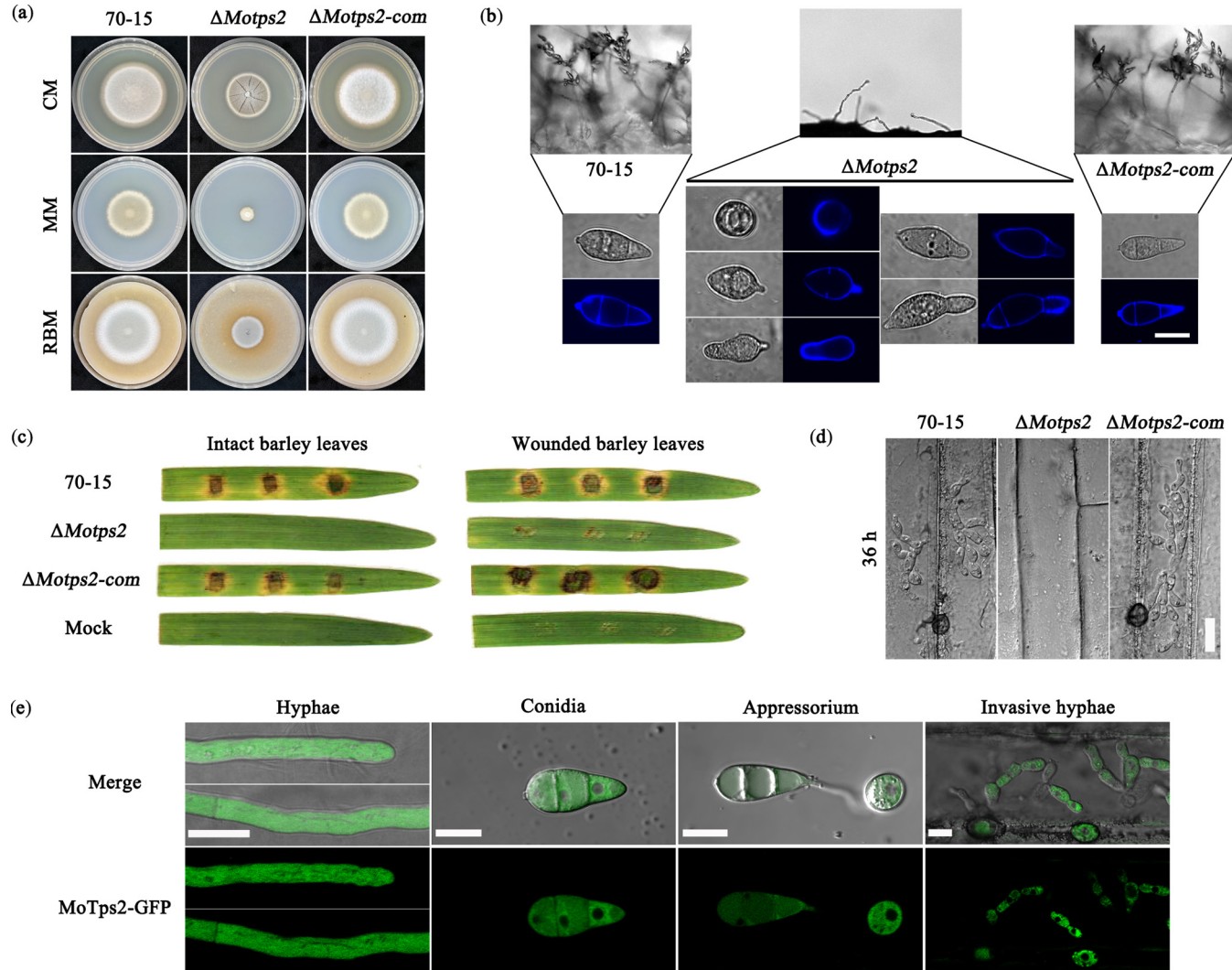

**FIG 1** MoTps2 is localized to cytoplasm and is important for growth, conidiation, and pathogenicity of *M. oryzae*. (a) MoTps2 is required for hyphal vegetative growth. Colonies of the wild type (70-15), Δ*Motps2* mutant, and complemented strain (Δ*Motps2*-com) were grown on complete medium (CM), minimal medium (MM), and rice bran medium (RBM) agar for 7 days. (b) MoTps2 is necessary for conidial development and morphology. Calcofluor white (CFW) was used to visualize conidial cell wall and septa, and most of the mutant conidia had abnormal shapes with only one or no septum. Bar = 10 $\mu$m. (c) The Δ*Motps2* mutant showed a defect in plant infection. Mycelial blocks of the various strains were inoculated on both intact and wounded barley leaves for 5 days. (d) MoTps2 is required for development of invasive hyphae. Hyphal blocks of the indicated strains were inoculated on barley leaves for 36 h and then observed under a microscope. Bar = 20 $\mu$m. (e) Confocal microscopic examination for localization of MoTps2-GFP fusion protein at different developmental stages. MoTps2 was observed to be expressed ubiquitously in the cytoplasm at all developmental stages of the fungus. Bar = 10 $\mu$m.

hpi (Fig. 2a and Table 1), suggesting that MoTps2 is important for appressorial development.

*M. oryzae* fungi normally form appressorium-like structures (ALSs) at their hyphal tips when inoculated on hydrophobic surfaces (25). To explore the role of MoTps2 in the formation of ALSs, mycelial balls of the wild type and Δ*Motps2* mutant were inoculated on hydrophobic glass, paraffin wax, and barley and rice leaves. Compared to the wild type, the Δ*Motps2* mutant was significantly reduced in the ability to form ALSs on the paraffin wax and barley and rice leaves and completely lost the ability to produce ALSs on the hydrophobic glass surface (Fig. 2b and c), suggesting that MoTps2 is required for sensing hydrophobicity signal. In addition, fewer and smaller ALSs were formed by the Δ*Motps2* mutant in comparison with the wild type (see Fig. S4 in the supplemental material). In *M. oryzae*, proper turgor pressure generated from accumulated glycerol in an appressorium is needed to rupture and penetrate the host cuticle, and this is indicated by the collapse of the appressorium (24, 26–28). To better understand the nonvirulence defect of the

**TABLE 1** Characterization of growth, conidiation, septum number, and appressorium formation in related strains[a]

| Strain | Vegetative growth (cm) on: | | | Conidiation (10⁴/ml)[b] | No. of septa[c] | | | Appressorium formation (%)[d] |
|---|---|---|---|---|---|---|---|---|
| | CM | MM | RBM | | 0 | 1 | 2 | |
| 70-15 | 4.26 ± 0.04 A | 3.17 ± 0.07 A | 4.24 ± 0.04 A | 94.9 ± 10.9 A | 4.0 ± 0.6 A | 11.0 ± 1.5 A | 85.0 ± 1.2 A | 93.0 ± 1.2 A |
| ΔMotps2 | 2.88 ± 0.09 B | 1.01 ± 0.07 B | 2.11 ± 0.38 B | 0.11 ± 0.33 B | 60.6 ± 2.4 B | 35.3 ± 2.4 B | 4.0 ± 1.2 B | 3.7 ± 0.9 B |
| ΔMotps2-com | 4.15 ± 0.04 A | 3.13 ± 0.08 A | 4.08 ± 0.07 A | 88.2 ± 8.97 A | 5.0 ± 0.6 A | 7.7 ± 1.9 A | 87.3 ± 1.5 A | 92.3 ± 1.9 A |

[a]Data are presented as the means ± SE of results from three independent experiments. Data were analyzed by unpaired two-tailed Student's $t$ test. Different letters indicate significant difference compared to the wild-type strain, 70-15 ($P < 0.01$).
[b]CM was used for the conidiation assay.
[c]Conidia of the indicated strains were harvested from 10-day CM plates.
[d]Shown is the percentage of appressorium formation on an artificial hydrophobic surface at 24 h postinoculation.

ΔMotps2 mutant, we assayed the turgor pressure in the ALSs formed by the wild type and ΔMotps2 mutant. Unlike the wild type, the ΔMotps2 mutant generated an excess level of turgor pressure, with only a few collapsed appressoria, even with increased supplementation of glycerol (Fig. 2d). Therefore, the abnormal turgor pressure generated by the ΔMotps2 mutant possibly led to its defect in plant infection. Taken together, we conclude here that MoTps2 is important for timely appressorial formation and generation of turgor pressure in *M. oryzae*.

**The ΔMotps2 mutant displays abnormal cell wall structure.** Our results indicated that the ΔMotps2 mutant has a defect in growth. Cell wall assembly has been shown to contribute to hyphal tip growth in filamentous fungi (29, 30). As such, we decided to check the effects of the cell-wall-perturbing agents calcofluor white (CFW) and Congo red (CR) on the growth rates of the various strains. The ΔMotps2 mutant had increased

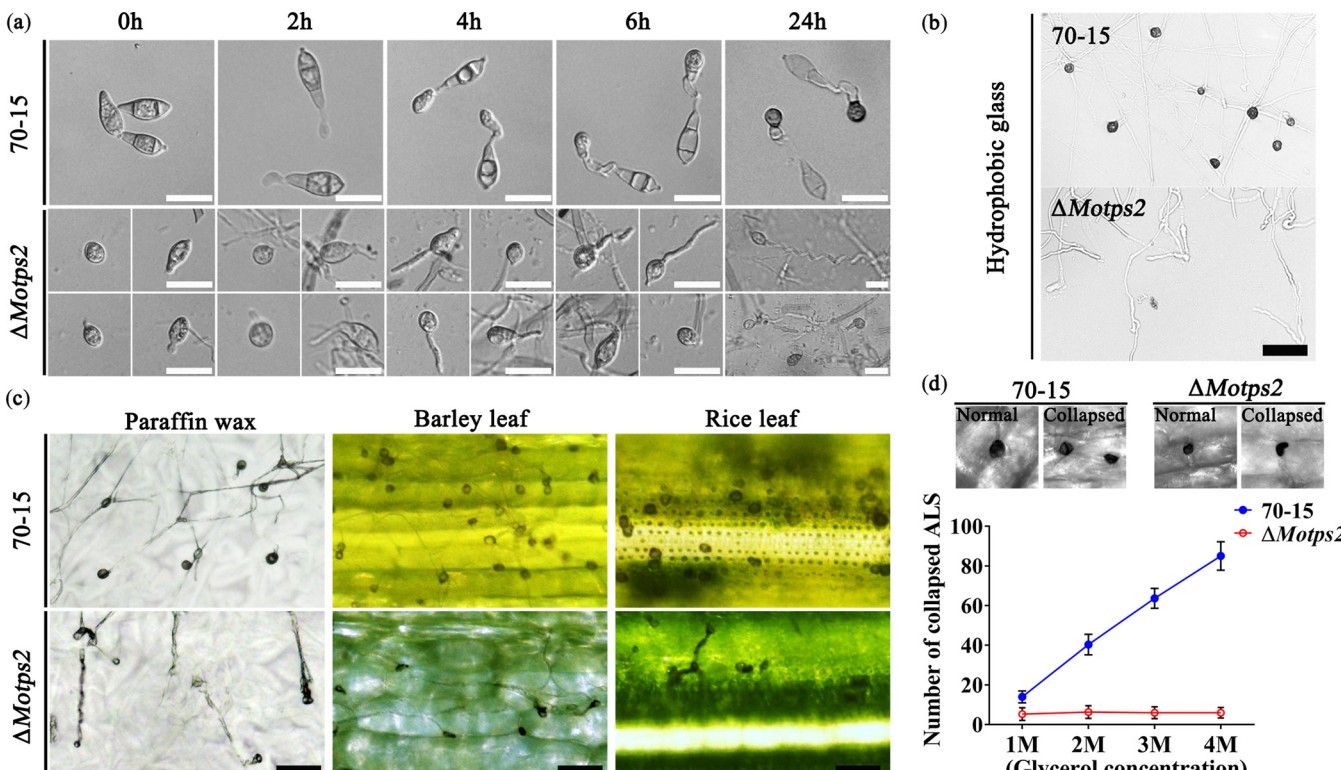

**FIG 2** MoTps2 is required for the formation of infection structures. (a) MoTps2 is essential for appressorial development. To induce appressorium formation, conidia of the wild type and ΔMotps2 mutant were harvested from 10-day-old CM cultures and then inoculated on hydrophobic surfaces for 2, 4, 6, and 24 h. The ΔMotps2 mutant shows a defect in appressorium formation at each time point. Bar = 20 μm. (b) The ΔMotps2 mutant failed to develop appressorium-like structures (ALSs) from hyphae on the artificial hydrophobic surface. Bar = 50 μm. (c) The number of appressorium-like structures formed by the ΔMotps2 mutant is significantly reduced on paraffin wax and barley and rice leaves. To induce ALS development, the fungal hyphae were inoculated on paraffin wax and barley and rice leaves for 3 days. Bar = 50 μm. (d) The ALS from the ΔMotps2 mutant generated abnormal turgor pressure compared to the wild type.

sensitivities to these compounds—more than those of the wild type and the complemented strain (Fig. 3a). Since chitin and glucan are the major components of the cell wall and CFW binds to nascent chitin and glucan (31, 32), we checked for possible deposition of the cell wall polymers in the wild type and ΔMotps2 mutant through CFW staining. Microscopic examination indicated that the ΔMotps2 mutant had excessive cell wall polymer deposition on the hyphae, especially at the hyphal tips (Fig. 3b). As chitin and glucans are both the major components of the fungal cell wall, we reasoned that the fungal cell wall structure could have likely been altered due to the deletion of MoTPS2. Further transmission electron microscopy (TEM) indicated that the cell wall of the ΔMotps2 mutant was obviously thicker than that of the wild type (Fig. 3c), likely due to the deposition of the cell wall polymers on the mutant's cell wall. To further verify this, we checked the transcription levels of some genes implicated in the maintenance of cell wall structure, including genes related to melanin (33), glucans (34, 35), and chitin (36–38), in both the wild type and ΔMotps2 mutant. The quantitative real-time PCR (qRT-PCR) results (see Fig. S5a in the supplemental material) suggest that the absence of MoTps2 impairs the expression of these genes, which may also have contributed to the abnormal cell wall structure.

The chitin synthase genes (CHS) are responsible for the production of chitin polymers, and CHS1, CHS3, and CHS4 genes have been shown to be involved in cell wall synthesis and maintenance of cell wall integrity in the filamentous fungus Neurospora crassa (31). In M. oryzae, MoCHS1 has been linked to appressorium formation and virulence, while MoCHS3 plays a role in cell wall integrity (36). Since the appropriate localization of CHS genes is essential for cell wall homeostasis (29, 39), and abnormal chitin deposition was detected in our ΔMotps2 mutant, we further checked the localizations of MoChs1 and MoChs3. Our results reveal that both MoChs1 and MoChs3 mainly localize to the Spitzenkörper in the wild type, while in the ΔMotps2 mutant, they are largely mislocalized to the plasma membrane and cytoplasm (Fig. 3d). This suggests a role played by MoTps2 in the proper localization of MoChs1 and MoChs3. Actin is an important component of the Spitzenkörper and is required for vesicle transport in filamentous fungi (29, 40, 41). To determine whether there is any link between actin organization and excessive cell wall polymer deposition on the hyphal tips, we treated the wild-type hyphae with the actin assembly inhibitor latrunculin A (Lat A) for 1 h and checked for changes in hyphal tip morphology and CFW staining. As shown in Fig. S5b, treatment with Lat A caused the morphology of the wild-type hyphal tips appear similar to that observed in the ΔMotps2 mutant hyphae (Fig. 3e). CFW staining of the treated hyphae also demonstrated accumulation of cell wall polymers along the cell walls. In addition, the Spitzenkörper localization of actin was impaired after Lat A treatment, suggesting a link between actin assembly and cell wall polymer deposition.

To understand the role of MoTps2 in actin assembly, we examined the localization of the Lifeactin-RFP (red fluorescent protein) fusion protein in both strains. An actin patch was found to accumulate at the Spitzenkörper and subapical collar in the wild type. In the ΔMotps2 mutant, the Spitzenkörper localization was impaired, and the actin patch failed to concentrate at the growing hyphal apex (Fig. 3g). In yeast, the polarisome scaffolding protein Spa2p interacts with an actin-interacting protein, Aip5, to regulate actin polymerization (42, 43). In M. oryzae, MoSpa2 is required for proper polarized growth (44). We therefore investigated the impact of MoTPS2 deletion on MoSpa2 localization by transforming the MoSpa2-GFP construct into the wild type and ΔMotps2 mutant. As expected, MoSpa2-GFP localized at the hyphal tip region in the wild type, while disruption of MoTPS2 obviously attenuates MoSpa2-GFP fluorescence intensity at the hyphal tip of the mutant (Fig. 3h). Consistent with these observations, the expression level of MoSpa2 was reduced in the ΔMotps2 mutant when assayed by Western blotting (Fig. 3h), which could have led to the abnormal actin polymerization. Overall, these results indicate that loss of MoTps2 leads to cytoskeleton disorganization and consequently alters the cell wall structure in M. oryzae.

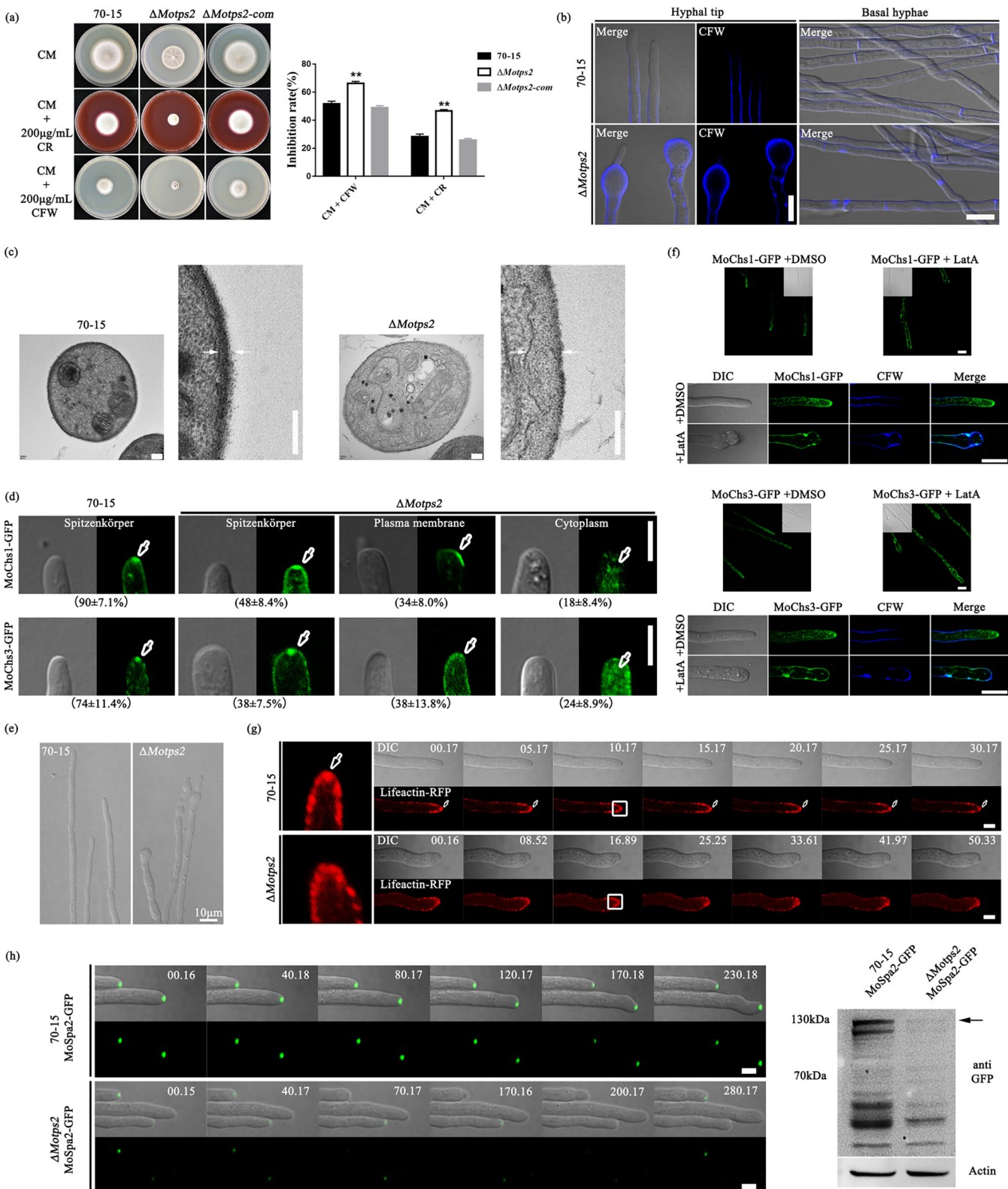

**FIG 3** Deletion of *MoTPS2* leads to abnormal cell wall structure. (a) The Δ*Motps2* mutants were more sensitive to calcofluor white (CFW) and Congo red (CR). The wild-type strain, Δ*Motps2* mutants, and Δ*Motps2*-com complemented strain were inoculated on CM containing the cell-wall-perturbing agents CFW and CR for 7 days. **, $P < 0.01$. (b) The Δ*Motps2* mutant displayed enhanced cell wall polymer deposition on hyphal tips and basal hyphae. Bar = 10 μm. (c) The Δ*Motps2* mutant displayed thicker cell wall structure compared to the wild type. To detect the cell wall structure of the wild type and Δ*Motps2* mutant, the strains were cultured in liquid CM for 2 days and then prepared for TEM examination. A pair of white arrows indicates cell wall size. (d) Disruption of *MoTPS2* alters the localization of MoChs1 and MoChs3. For confocal examination, the various strains were grown on CM for 7 days; hyphal

**The ∆*Motps2* mutant undergoes spontaneous mutations to restore its defects.** We observed that the ∆*Motps2* mutant was not very stable and eventually produces spontaneous suppressor strains with a higher growth rate than the mutant (see Fig. S6a in the supplemental material). We isolated these strains and further investigated their phenotypes. The spontaneous suppressor strain (presented here as ∆*Motps2-m*) was partially rescued in the defects of the ∆*Motps2* mutant in growth, hyphal morphology, conidiophore formation, and pathogenicity. Moreover, TEM examination revealed that the ∆*Motps2-m* strain had a similar cell wall thickness to the wild type (Fig. 4a and Table 2).

We speculated that the ∆*Motps2-m* spontaneous suppressor strains could be restored in actin organization and distribution of chitin synthase-related proteins, thereby rescuing ∆*Motps2* defects. To validate this hypothesis, we also expressed the Lifeactin-RFP, MoChs1-GFP, MoChs3-GFP, and MoSpa2-GFP fusion proteins in ∆*Motps2-m* strains. Confocal microscopic examination showed that the localizations of actin and chitin synthase-related proteins in the ∆*Motps2-m* strain are similar to those in the wild type. In addition, the localization of MoSpa2 is also rescued in the ∆*Motps2-m* mutant (Fig. 4b).

**Loss of MoTps3 activity partially rescues ∆*Motps2* defects.** To better understand the mechanism of the spontaneous suppression, two independent spontaneous strains were isolated, and their genomic DNAs were extracted for whole-genome sequence analysis. By so doing, suppressor mutations were identified in MGG_11098 (base transition), MGG_05689 (base transition), MGG_07789 (base transition), MGG_08555 (base transversion), MGG_01956 (base transition), MGG_03973 (base transversion), and MGG_14118 (base deletion and base insertion) (Fig. S6b). We noticed that both spontaneous suppressor strains had a frameshift mutation in the open reading frame of their *MoTPS3* genes (MGG_14118), the regulatory subunit of the TPS complex. The spontaneous mutant strains have 2 base deletions at $C^{1930}$ and $G^{1931}$ of the GT20-TPS (glycosyltransferase family 20) domain and 1 base insertion (base A) between $A^{3139}$ and $T^{3140}$ of the OtsB (trehalose 6-phosphatase) domain of the *MoTPS3* open reading frame, which could have led to the loss of MoTps3 activity (Fig. 4c; Fig. S6b). To confirm this result, we further generated ∆*Motps2*∆*Motps3* double deletion mutants. Phenotype assays suggested that loss of MoTps3 partially rescues the ∆*Motps2* defects, including vegetative growth, hyphal morphology, conidiogenesis, and virulence on plants (Fig. 4d and Table 2).

In yeast, the regulatory subunits of the TPS complex, Tsl1p and Tps3p, are responsible for the proper homeostasis of the trehalose biosynthesis process (8). In *M. oryzae*, MoTps3 is the only regulatory subunit of the TPS complex, and previous studies have indicated that MoTps3 can regulate MoTps1 activity (12). It is possible that loss of MoTps3 activity has effects on the production of T6P and trehalose. When assayed by gas chromatography-mass spectrometry (GC-MS), the loss of MoTps2 significantly increased the levels of T6P, with reduced trehalose biosynthesis, compared to the wild type. On the other hand, the ∆*Motps2-m* and ∆*Motps2*∆*Motps3* mutants showed lower levels of both trehalose and T6P compared to the ∆*Motps2* mutant (Fig. 4e). Overall, these results imply that MoTps3 is a spontaneous suppressor of MoTps2, and loss of MoTps3 activity partially rescues ∆*Motps2* defects in *M. oryzae*.

**FIG 3** Legend (Continued)
blocks were used for further microscopic examinations. Bar = 5 $\mu$m. The means ± standard deviation (SD) were calculated based on three independent experiments by measuring 30 hyphae in each replicate. (e) The ∆*Motps2* mutant showed abnormal hyphal morphology. The wild type and ∆*Motps2* mutant were grown on CM for 7 days; swollen and uneven structures were detected at the hyphal tips of the ∆*Motps2* mutant. (f) Localization of MoChs1 and MoChs3 is actin dependent. Strains expressing MoChs1-GFP and MoChs3-GFP were cultured on solid CM for 7 days. Lat A was used at a final concentration of 5 $\mu$g/ml. Bar = 10 $\mu$m. The Spitzenkörper localizations of MoChs1 and MoChs3 were impaired after treatment with Lat A. (g) The ∆*Motps2* mutant showed abnormal hyphal morphology, and loss of MoTps2 impairs the organization of actin. The Lifeactin-RFP localizes to the Spitzenkörper (marked by white arrows) in the wild type, while it escapes to the membrane in the ∆*Motps2* mutant. Bar = 5 $\mu$m. (h) Deletion of *MoTPS2* impairs the expression of MoSpa2-GFP in growing hyphae. The expression level of MoSpa2 is reduced in the ∆*Motps2* mutant. Strains expressing MoSpa2-GFP were cultured in CM for 2 days and then used for a further Western blot assay. The MoSpa2-GFP band (marked by a black arrow) was obviously reduced in the ∆*Motps2* mutant compared to the wild type (70-15). Bar = 5 $\mu$m.

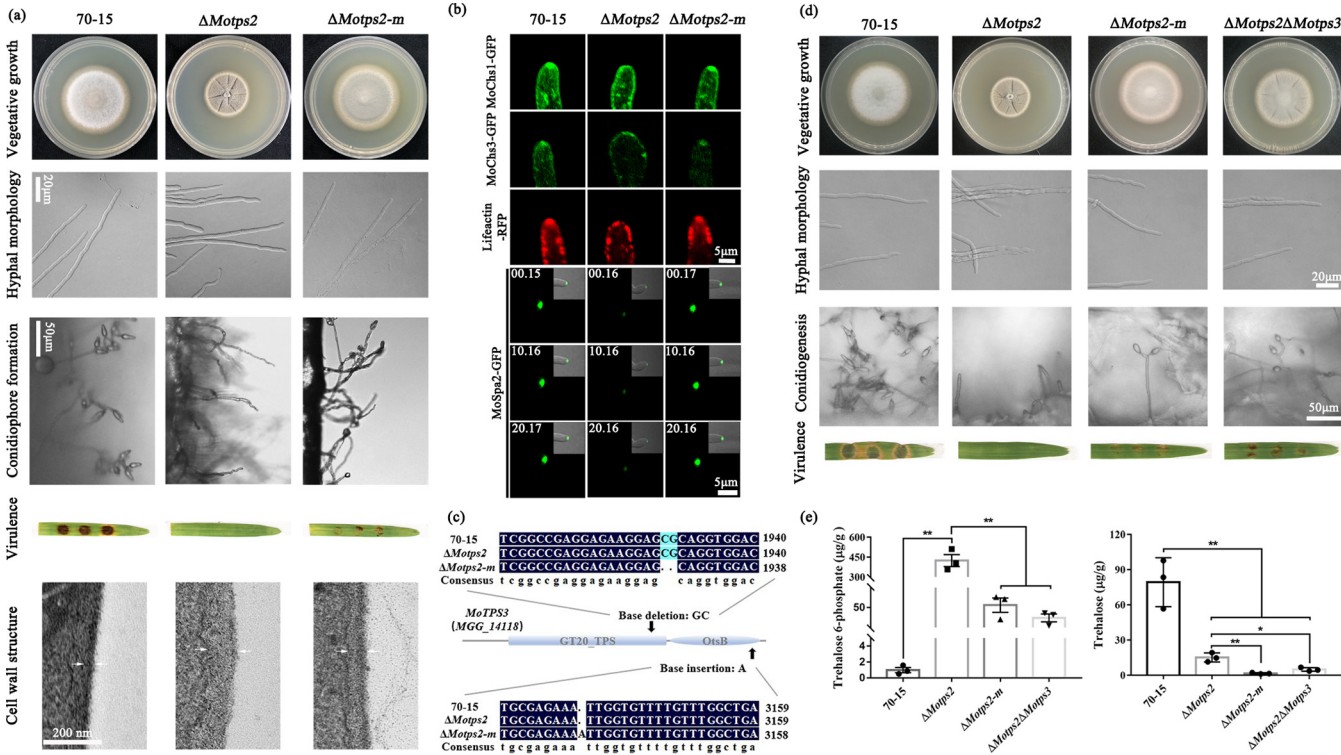

**FIG 4** Mutation in *MoTPS3* partially rescues Δ*Motps2* defects. (a) Phenotype assays of a spontaneous suppressor strain (Δ*Motps2-m*). The Δ*Motps2-m* spontaneous suppressor strain is partially restored in the defects observed in Δ*Motps2* mutant (defects in growth, hyphal morphology, conidiogenesis, pathogenicity, and cell wall structure). (b) Localization of chitin synthase and cytoskeleton-related proteins. Strains expressing MoChs1-GFP, MoChs3-GFP, Lifeactin-RFP, and MoSpa2-GFP fusion proteins were grown on CM for 7 days; hyphae from the cultures were used for further confocal examinations. (c) Whole-genome sequencing analysis of the Δ*Motps2-m* spontaneous mutation strain. A frameshift mutation was detected in the coding region of *MoTPS3*. (d) Phenotype assays of the Δ*Motps2*Δ*Motps3* mutant. Loss of MoTps3 can also recover defects of Δ*Motps2* mutant. (e) Quantitative analysis of the trehalose and trehalose 6-phosphate contents. The indicated strains were cultured in liquid CM at 28°C at 110 rpm for 3 days and then collected for further GC-MS analysis. *, $P < 0.05$; **, $P < 0.01$. A pair of white arrows indicates cell wall size.

**Deletion of *MoTPS1* partially rescues the defects of the Δ*Motps2* mutant.** To further characterize the suppression mechanism of MoTps3, yeast two-hybrid and bimolecular fluorescence complementation (BiFC) assays were carried to identify the various interactions among the TPS complex subunits. As shown in Fig. 5a and b, MoTps1 physically interacts with MoTps2 and MoTps3, while no direct interaction was detected between MoTps2 and MoTps3. We therefore hypothesize that MoTps3 partially rescues Δ*Motps2* defects by regulating MoTps1 activity. To validate this hypothesis, we generated the Δ*Motps1* mutant and Δ*Motps2*Δ*Motps1* double deletion mutant, which were verified by PCR and Southern blot assays (Fig. S2b and c). Phenotype assays indicated that deletion of *MoTPS1* partially recovered the defects of the Δ*Motps2* mutant in hyphal growth, conidiogenesis, virulence, and cell wall polymer deposition (Fig. 5c and Table 2). Moreover, TEM examination revealed that the cell wall thicknesses of the Δ*Motps2-m* mutant and the Δ*Motps2*Δ*Motps1* double deletion mutant were similar to that of the wild type (Fig. 5d and Table 2). Since a previous study indicated that loss of MoTps3 impairs MoTps1 activity (12), we conclude that mutation in *MoTPS3* (due to *MoTPS2* deletion) suppresses MoTps1 activity and therefore partially restores Δ*Motps2* defects in growth, conidiogenesis, cell wall structure, and pathogenicity.

**Trehalose 6-phosphate accumulation is responsible for the growth, conidiogenesis, and pathogenicity defects in the Δ*Motps2* mutant.** To further understand why deletion of *MoTPS1* or *MoTPS3* in the Δ*Motps2* mutant can partially rescue its defects, we then assayed T6P and trehalose production. The key metabolites in the TPS complex pathway are trehalose and trehalose 6-phosphate. If the low level of the final product (trehalose) is the major reason that led to the phenotypic defects of the Δ*Motps2* mutant,

**TABLE 2** Comparison of growth, conidiation, cell wall thickness, and virulence of the related strains[a]

| Parameter | Result for strain[a] | | | | | | | |
|---|---|---|---|---|---|---|---|---|
| | 70-15 | ΔMotps2 | ΔMotps2-m | ΔMotps2ΔMotps3 | ΔMotps1 | ΔMotps2ΔMotps1 | 70-15-TreC-1 | ΔMotps2-TreC-1 |
| Mycelial growth (cm)[b] | 4.38 ± 0.05 A | 2.81 ± 0.04 B | 4.10 ± 0.03 A | 3.97 ± 0.02 A | 4.31 ± 0.05 B | 4.07 ± 0.05 B | 4.34 ± 0.04 B | 4.29 ± 0.02 B |
| Conidiation ($10^4$/ml)[b] | 107.02 ± 4.96 A | 0.20 ± 0.13 B | 18.20 ± 1.89 B | 10.18 ± 1.95 B | 8.80 ± 1.12 B | 13.10 ± 0.73 B | 106.00 ± 1.17 A | 16.61 ± 6.5 B |
| Cell wall thickness (nm)[c] | 64.22 ± 1.45 A | 117.89 ± 3.91 B | 60.07 ± 1.83 A | ND | 64.77 ± 2.81 A | 68.53 ± 2.47 A | ND | ND |
| Lesion level[d] | +++ | = | + | + | + | + | +++ | + |

[a]Data are presented as the means ± SE of results from three independent experiments. Data were analyzed by unpaired two-tailed Student's t test. Different letters indicate significant difference compared to the wild-type strain, 70-15 ($P < 0.01$).

[b]CM was used for growth and conidiation assays.

[c]ND, no detection.

[d]+++, similar lesion level compared to the wild type; +, reduced lesion level compared to the wild type; =, failed to produce lesion.

mSystems®

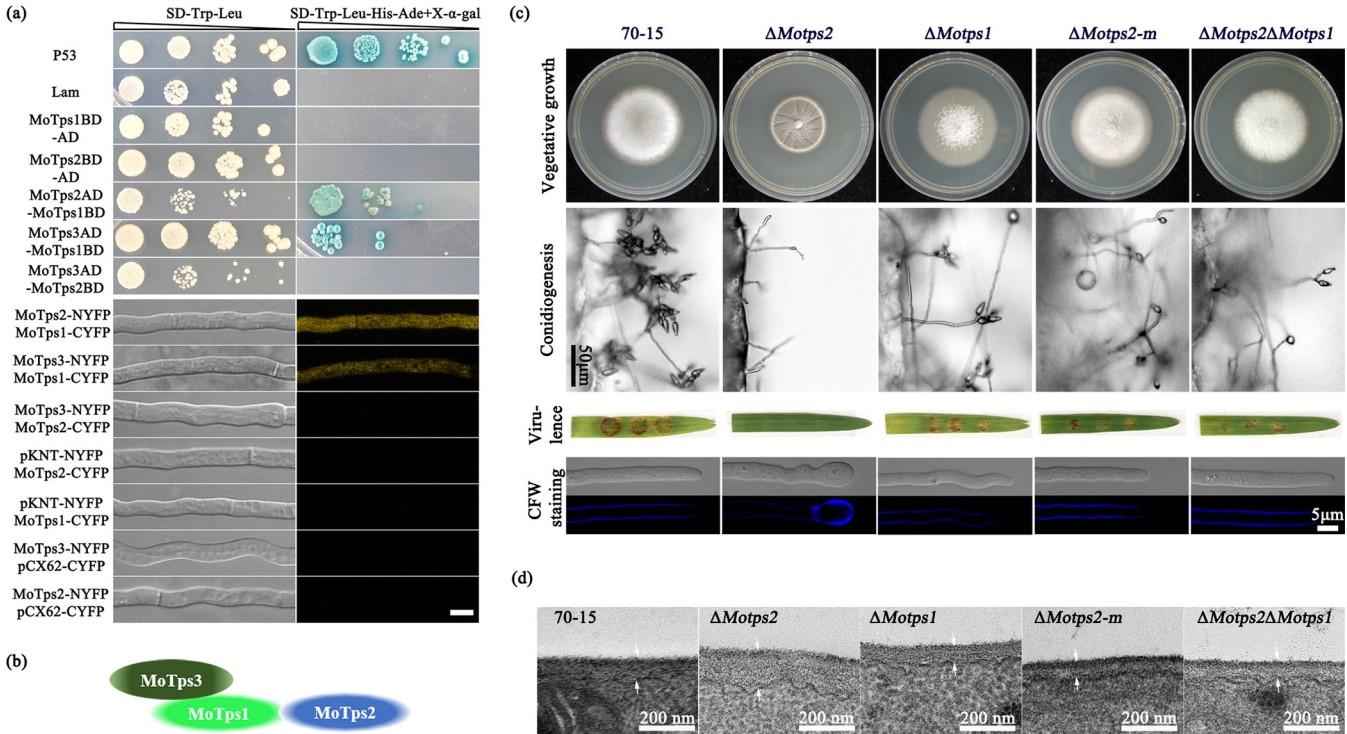

**FIG 5** Deletion of *MoTPS1* from the Δ*Motps2* mutant partially rescues Δ*Motps2* defects. (a) Yeast two-hybrid and BiFC assays to validate the interaction of the TPS complex. MoTps1 physically interacts with MoTps2 and MoTps3; no direct interaction between MoTps2 and MoTps3 was detected through yeast two-hybrid and BiFC assays. Bar = 5 $\mu$m. (b) Schematic diagram of the interaction of the TPS complex. (c) Phenotype assays of the Δ*Motps2*Δ*Motps1* double deletion mutant. The Δ*Motps2*Δ*Motps1* double deletion mutant has similar phenotypes to the Δ*Motps2-m* spontaneous suppressor strain, including mycelial growth, conidiophore formation, pathogenicity, and cell wall polymer deposition. (d) Cell wall structures of the indicated strains. A pair of white arrows indicates cell wall size.

then the Δ*Motps1* mutant should have more severe defects than the Δ*Motps2* strain, since the Δ*Motps1* strain has less trehalose than the Δ*Motps2* strain (Fig. 6a). However, our data supported that the defects in the Δ*Motps1* mutant are not as severe as those in the Δ*Motps2* mutant, and double deletion of *MoTPS1* and *MoTPS3* can partially rescue Δ*Motps2* defects (Fig. 4c and Fig. 5c). In addition, exogenous supplementation of trehalose failed to restore the Δ*Motps2* defects (see Fig. S7a in the supplemental material), indicating that the reduced level of trehalose is not the primary reason for Δ*Motps2* spontaneous mutation. As shown in Fig. 6a, the T6P content of the Δ*Motps2* mutant was increased by over 800-fold compared to the wild type, while double deletion of *MoTPS1* and *MoTPS2* significantly reduced the T6P content compared to that of the Δ*Motps2* mutant. It is likely that the loss of MoTps2 leads to accumulation of T6P, which in turn impairs growth, conidiogenesis, and pathogenicity in *M. oryzae*. To test this hypothesis, we obtained trehalose 6-phosphate hydrolase, TreC (which specifically hydrolyzes trehalose 6-phosphate to glucose and glucose-6-phosphate) (45), from *Escherichia coli* (Fig. 6b) and generated a TreC-GFP fusion protein. TreC-GFP was then transformed into the wild type and Δ*Motps2* mutant (Fig. 6c). Microscopic examinations of TreC-GFP indicated that the fusion protein localizes to the cytoplasm in both the wild type and Δ*Motps2* mutant (Fig. 6d). Further phenotype assays showed that the expression of TreC partially restores the Δ*Motps2* defects (Fig. 6e). In addition, expression of TreC significantly reduced the T6P level in the Δ*Motps2* mutant (Fig. 6a). Coupled with the results observed in the Δ*Motps2-m* and Δ*Motps2*Δ*Motps3* mutants (Fig. 4e), we conclude that the spontaneous mutation of Δ*Motps2* was mainly due to the accumulation of T6P in the mutant.

To gain more insights into the effects of trehalose 6-phosphate accumulation in the fungus, the wild type was cultured on CM plates with or without trehalose 6-

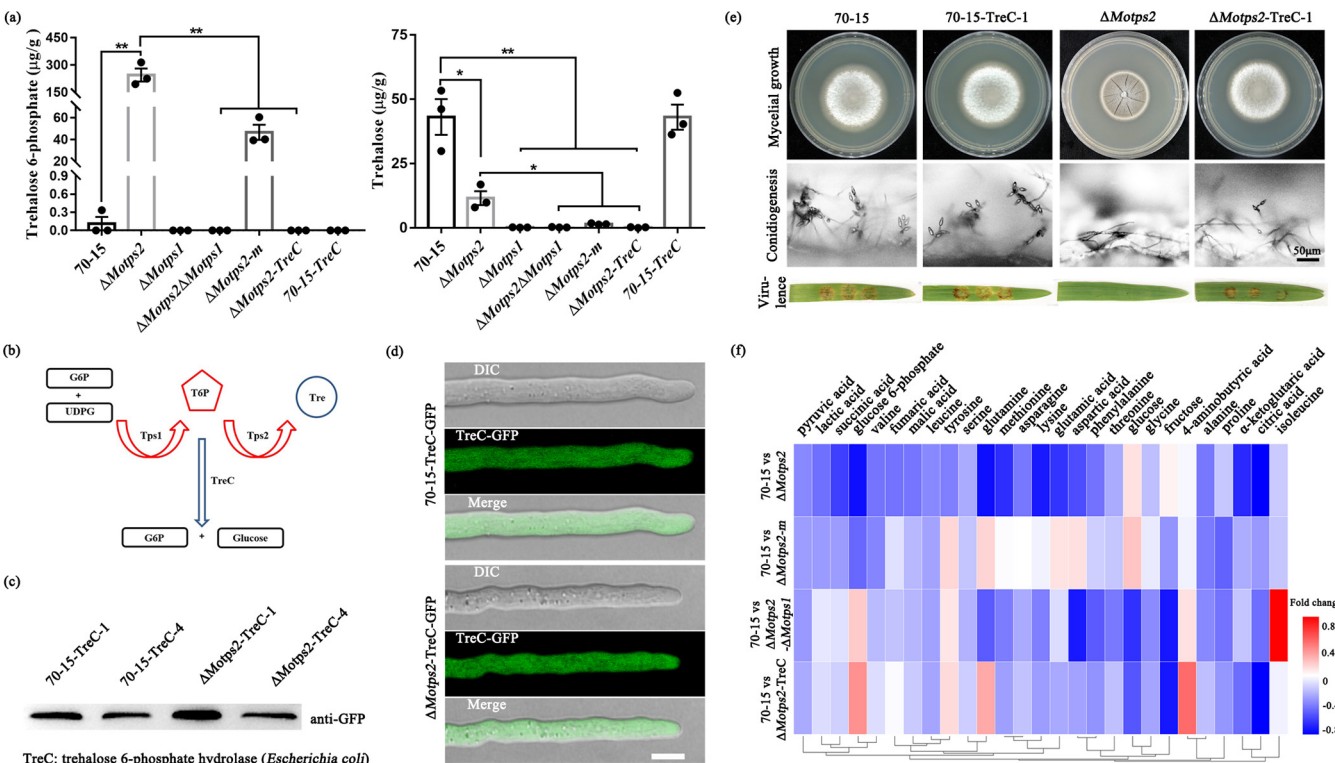

**FIG 6** Heterologous expression of trehalose 6-phosphate hydrolase TreC partially restores Δ*Motps2* defects. (a) Trehalose 6-phosphate and trehalose contents in related strains. *, *P* < 0.05; **, *P* < 0.01. (b) Trehalose 6-phosphate hydrolase TreC can specifically bind to trehalose 6-phosphate and catalyze the hydrolysis of trehalose 6-phosphate into glucose and glucose-6-phosphate. (c) Western blot assay confirming the expression of trehalose 6-phosphate hydrolase in the wild type and Δ*Motps2* mutant. (d) Subcellular localization of TreC-GFP in wild-type strain 70-15 and the Δ*Motps2* mutant. Bar = 5 μm. (e) Phenotype assays of the strains expressing the trehalose 6-phosphate hydrolase TreC. Expression of TreC partially restores Δ*Motps2* mutant defects in growth, hyphal morphology, conidiogenesis, and pathogenicity. (f) Summary of the metabolite change among the wild-type and Δ*Motps2*, Δ*Motps2-m*, Δ*Motps2*Δ*Motps1*, and Δ*Motps2*-TreC mutant strains.

phosphate. The results indicate that exogenous T6P has no obvious effect on the vegetative growth of the fungus (Fig. S7b), most likely due to the presence of MoTps2, which dephosphorylates the T6P to trehalose, or the exogenous T6P is not efficiently permeable into the fungal cell wall, as previously reported in plants (46, 47). In addition, no clear difference was observed in conidial germination and appressorial development (Fig. S7b). Based on these findings, we conclude that the intracellular accumulation of T6P is responsible for the defects in growth, conidiogenesis, and pathogenicity observed in the Δ*Motps2* mutant.

It has been reported that maintaining amino acid homeostasis is important for normal growth and pathogenicity of *M. oryzae* (48–55). In the Δ*Motps2* mutant, the intracellular levels of 11 amino acids (alanine, valine, leucine, methionine, glutamic acid, phenylalanine, asparagine, glutamine, lysine, tyrosine, and aspartic acid) in vegetative hyphae were significantly lower than those of the wild type (Fig. 6f). In addition, the Δ*Motps2* mutant has lower levels of metabolites that are involved in glycolysis (glucose-6-phosphate, pyruvic acid, and lactic acid) and the TCA cycle (succinic acid, fumaric acid, malic acid, α-ketoglutaric acid, and citric acid) compared to the wild type. To better understand the relationship between *MoTPS2* deletion and the metabolites' abundance, we checked the primary metabolite levels in the Δ*Motps2*-m, Δ*Motps2*Δ*Motps1*, and Δ*Motps2*-TreC strains. Interestingly, most of the metabolite levels were restored when the intracellular T6P level was reduced (by *MoTPS3* mutations, *MoTPS1* deletion, or TreC expression) (Fig. 6a and f), suggesting that the intracellular balance of T6P is required for the homeostatic regulation of these metabolites.

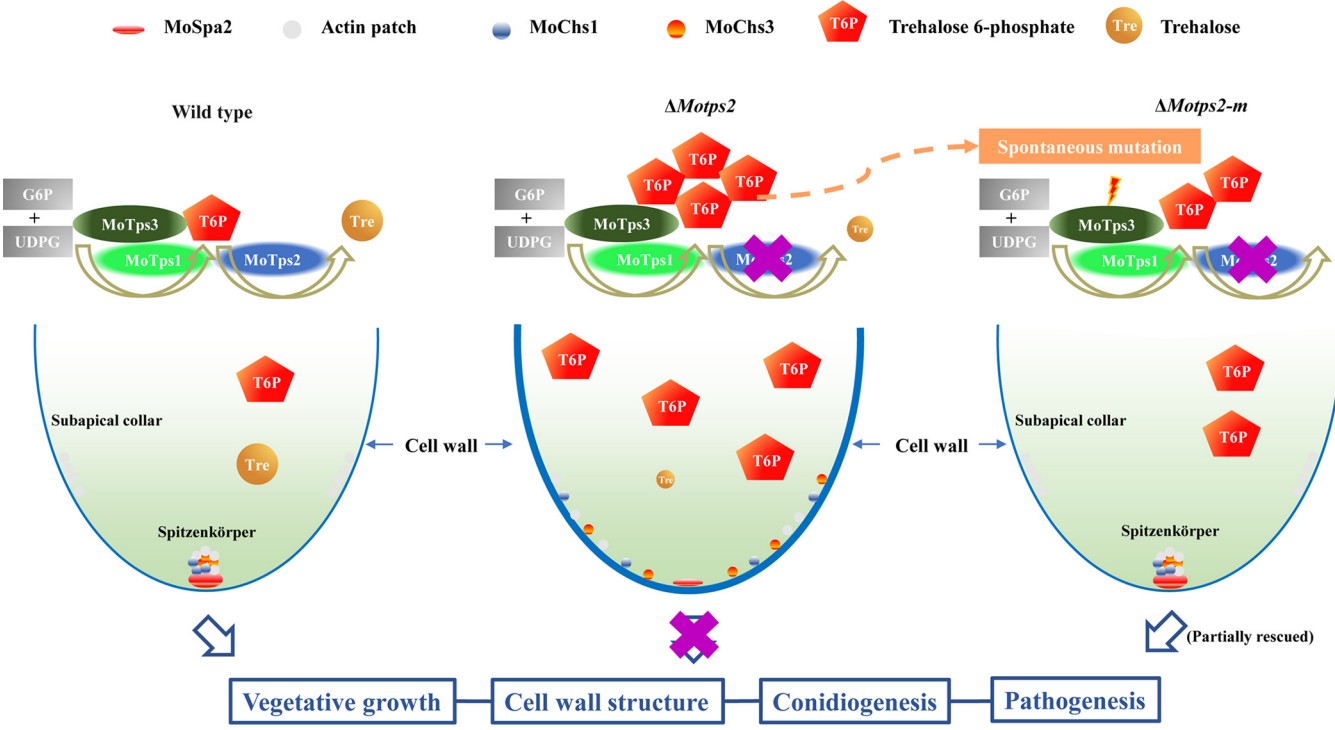

**FIG 7** A proposed model for the regulation of trehalose 6-phosphate homeostasis in *M. oryzae*. In the wild type, MoTps1 converts glucose 6-phosphate (G6P) and UDP-glucose (UDPG) into trehalose 6-phosphate, which is then dephosphorylated by MoTps2 to form trehalose. MoTps3, the regulatory subunit of the TPS complex, couples with MoTps1 and MoTps2 to modulate the intracellular homeostasis of trehalose 6-phosphate and therefore ensure proper fungal development and pathogenicity. In the ΔMotps2 mutant, loss of MoTps2 activity leads to accumulation of T6P, which in turn impairs the actin organization and expression of MoSpa2, resulting in defects in tip growth. Moreover, the disorganization of the actin skeleton leads to mislocalization of chitin synthase as well as abnormal cell wall structure. In the ΔMotps2-m mutant, the high level of T6P possibly induces the spontaneous mutation in *MoTPS3*, which in turn suppresses MoTps1 activity and therefore reduces the intracellular level of trehalose 6-phosphate, which as a consequence, partially recovers the ΔMotps2 mutant's defects.

## DISCUSSION

Trehalose 6-phosphate, the intermediate of the trehalose biosynthesis pathway, regulates metabolic pathways in yeast and plants (56–58). The cytotoxic effects of T6P accumulation have been established in different species (16, 21–23, 59, 60). In this study, we investigated the role of the TPS complex subunit MoTps2 in *M. oryzae*, the causal agent of the rice blast disease, and uncovered a spontaneous correction way of the TPS complex to modulate T6P homeostasis (Fig. 7). Unlike in the wild type, deletion of *MoTPS2* significantly increased the level of T6P, which results in the disorganization of cytoskeleton, and thus impairs the distribution of cell wall polymers, leading to defects in the growth, conidiogenesis, and pathogenicity of the fungus. To rebalance the intracellular level of T6P, spontaneous frameshift mutations occurred in *MoTPS3* to repress MoTps1 activity, resulting in a decrease and rebalance in the intracellular level of T6P in the ΔMotps2 mutant. These spontaneous mutations also recovered the normal organization of the cytoskeleton as well as the distribution of chitin synthases, thereby partially restoring the ΔMotps2 defects. To our knowledge, this is the first time to uncover how the TPS complex through spontaneous mutation in its regulatory subunit modulates T6P homeostasis in eukaryotes.

Fungal cell walls are composed of different components, including chitin, glucans, glycoproteins, and melanin (31, 61). Chitin, one of the major components of cell wall, is synthesized by different chitin synthase (*CHS*) enzymes. The chitin synthases are divided into seven families in most filamentous fungi, and most of them have been shown to localize at the hyphal apex (62, 63). Following their synthesis in the ribosomes, chitin synthases are packaged in transport vesicles and delivered to the plasma membrane for cell wall biosynthesis, and cytoskeleton organization is essential for the

transport of these vesicles (29, 40, 41). Studies with different species indicated that deletion of Tps2 impairs the cell wall integrity pathway and also has an effect on chitin synthesis (16, 19, 22, 64), but the mechanistic relationship remains unclear. In this study, we found that loss of MoTps2 impairs actin organization, which leads to mislocalization of *CHS* enzymes. In addition, mislocalization of *CHS* results in abnormal chitin deposition, which alters the cell wall structure (Fig. 3c; Fig. S5b). In fungi, there is no previous report that links the level of intracellular T6P to cytoskeleton organization. In plants, F-actin and microtubule are the key structures that determine cell shape, and T6P level has been shown to affect cell morphogenesis (56, 65). It is likely that the defect observed in the Δ*Motps2* mutant in cytoskeleton organization is related to its elevated intracellular T6P level. Consistent with this, we showed that the mislocalizations of MoSpa2 and actin were restored when the intracellular level of T6P was reduced (Fig. 4b). Previous studies revealed that cell metabolism processes provide energy to support cytoskeleton rearrangement (66). Herein, our results indicate that loss of MoTps2 impairs metabolite homeostasis *in vivo*. The intracellular abundances of some metabolites involved in glycolysis and the TCA cycle, as well as the abundances of some important amino acids, were significantly low in the Δ*Motps2* mutant. Interestingly, most of the metabolite changes were restored when the T6P level was reduced (Fig. 6; see Table S3 in the supplemental material). This clearly demonstrates that T6P homeostasis is crucial for normal physiologic balance of these metabolites. We therefore speculate that the increased T6P level (due to loss of MoTps2) impairs the fungal energy metabolism *in vivo*, and as a consequence, not enough energy is provided for cytoskeletal remodeling. In this situation, the cytoskeleton organization is impaired, leading to abnormal cell wall polymer distribution and thus altering the cell wall structure.

The most widely distributed trehalose biosynthesis pathway contains two key enzymes—trehalose 6-phosphate synthase (Tps1) and trehalose 6-phosphate phosphatase (Tps2)—in a complex called the TPS complex (7, 67). In yeast, the TPS complex contains two regulatory subunits, named Tps3p and Tsl1p. Both Tps3p and Tsl1p physically interact with Tps1p and Tps2p. In addition, Tps1p and Tps2p strongly interact with each other, while Tps3p and Tsl1p do not interact under normal conditions (8, 68). Unlike in yeast, the TPS complex in *M. oryzae* is composed of only three subunits, namely, MoTps1, MoTps2, and MoTps3, which serve as the regulatory subunit. Furthermore, we did not detect any interaction between MoTps2 and MoTps3; MoTps1, however, interacts with both MoTps2 and MoTps3 in vegetative hyphae under normal growth conditions (Fig. 5a). This suggests that the regulatory function of the TPS complex in *M. oryzae* differs from that in yeast. We showed that disruption or deletion of *MoTPS3* in the Δ*Motps2* mutant reduces the intracellular level of T6P and therefore suppresses the defects of the Δ*Motps2* mutant (Fig. 4d and e). Furthermore, we found that deletion of *MoTPS1* in the Δ*Motps2* mutant also partially rescues its defects (Fig. 5c and d). Since T6P is an inhibitor of Tps1p, and in plants, T6P serves as a signaling molecule (3, 69, 70), we speculate that the T6P level must reach a threshold to inhibit MoTps1 activity. At a persistent threshold level, spontaneous mutation of *MoTPS3* is induced (likely via the T6P signaling), which in turn suppresses MoTps1 activity to reduce the level of T6P. It will be interesting to understand the underlying mechanism of how MoTps3 senses the accumulation of intracellular T6P and induces self-mutation to compromise the defects caused by T6P accumulation.

The trehalose biosynthesis pathway widely exists in fungi and has been considered a potential antifungal target for the development of novel antifungal drugs (2, 3, 69). Our results indicated that T6P accumulation, caused by the deletion of *MoTPS2*, has toxic effects on the cell and results in growth, conidiation, and pathogenicity defects. Nevertheless, *M. oryzae* can alleviate these defects through induction of spontaneous mutation of the *MoTPS3* gene. Our results suggest that the spontaneous mutation of *MoTPS3* further suppresses MoTps1 activity. Under this condition, the biological

functions of the TPS complex subunits are almost suppressed. However, the spontaneous mutation strain has no significant difference in growth compared to the wild type and can also produce conidia and pathogenicity lesions (Fig. 4). In addition to the spontaneous mutation identified in the *MoTPS3* gene sequence, whole-genome sequencing analysis also revealed six other mutation sites that exist in the spontaneous mutant strains (Fig. S6b), suggesting that *M. oryzae* has multiple ways of overcoming the effects of T6P accumulation. The implication of this is that the targeting of TPS/TPP trehalose pathway for development of fungicides in the management of rice blast disease needs to be assessed carefully. Moreover, investigating the possibility of spontaneous mutations when the TPS/TPP trehalose biosynthesis pathway is targeted in other fungal species will also be useful.

## MATERIALS AND METHODS

**Fungal strains and culture conditions.** *M. oryzae* strain 70-15 was used as the wild type in this study; all other mutants used in this study were derived from 70-15, as listed in Table S1 in the supplemental material. For growth assays, hyphal blocks of the various strains were inoculated on complete medium (CM) (71), minimal medium (MM) (13), and rice bran medium (RBM) (72) plates at 28°C in the dark for 7 days. Conidiation assays were performed on CM agar plates at 28°C with periodic exposure to 12-h light/dark cycles for 7 days. To test for sensitivity to cell-wall-perturbing agents, the mycelial plugs were placed on CM agar plates containing 200 $\mu$g/ml calcofluor white (CFW) (fluorescent brightener 28; Sigma) and 200 $\mu$g/ml Congo red (CR) (Sigma-Aldrich) for 7 days.

**Development of infectious structures and pathogenicity assay.** To induce the formation of appressoria, 20-$\mu$l conidial suspensions (1 × 10$^4$ cells/ml) from the various strains were placed on hydrophobic coverslips for 2, 4, 6, and 24 h and finally observed under a microscope. For development of appressorium-like structures, mycelial balls were harvested from 2-day-old CM agar cultures and then placed on hydrophobic coverslips, paraffin wax, and barley and rice leaves for 72 h, respectively. For the penetration assay, hyphal blocks of the indicated strains were placed on 7-day-old barley leaves, incubated at 28°C for 36 h, and then observed under a confocal microscope. For the plant infection assay, mycelial plugs were placed on intact or wounded barley leaves and incubated in a humid chamber at 28°C for 5 days.

**Generation of gene deletion mutants and complementation.** For generation of gene deletion mutants, a split-marker approach was used (73). For generation of *MoTPS1* and *MoTPS2* deletion mutants, ~1-kb upstream and downstream sequences of *MoTPS1* and *MoTPS2* genes were amplified using specific primers (see Table S2 in the supplemental material). The resulting fragments were ligated with a previously amplified complete sequence of a hygromycin resistance gene cassette at its flanking ends using splicing by overlap extension (SOE) PCR, and the product was finally transformed into the protoplasts of the wild-type strain, 70-15. For generation of the Δ*Motps2*Δ*Motps1* and Δ*Motps2*Δ*Motps3* double deletion mutants, ~1-kb upstream and downstream sequences of *MoTPS1* and *MoTPS3* genes were amplified, and the resulting fragments were ligated with amplified complete sequence of the neomycin resistance gene cassette and then transformed into the protoplasts of the Δ*Motps2* mutant. The transformants were screened by PCR using specific primers (Table S2) and further confirmed by Southern blotting using the DIG-High Prime DNA labeling and detection starter kit I (Roche). For Δ*Motps2* mutant complementation, the coding region and native promoter of the *MoTPS2* gene were amplified using specific primers (Table S2) and cloned into a pKNT-GFP vector using the ClonExpress II One Step cloning kit (Vazyme). Subsequently, the resulting MoTps2-GFP fusion protein was transformed into the Δ*Motps2* mutant protoplast and further verified by PCR screening and confocal microscopic examination for GFP detection.

**Construction of MoChs1-GFP, MoChs3-GFP, and TreC-GFP fusion proteins.** For construction of MoChs1-GFP and MoChs3-GFP vectors, the various genes encoding the *M. oryzae* proteins were amplified using specific primer pairs (Table S2) and then cloned into a pKNT-GFP vector, respectively. For the construction of TreC-GFP fusion protein, the coding sequence of *Escherichia coli* trehalose 6-phosphate hydrolase (TreC) was amplified and cloned into a pKNT-RP27-GFP vector. The resulting fusion vectors were further verified by sequence analysis (Sangon, Shanghai, China).

**Microscopic examination.** For observation of hyphal morphology, the various strains were cultured on CM agar plates for 7 days, after which hyphal blocks were placed upside down on coverslips and used for further confocal microscopic examinations. For staining of the chitin component of the cell wall, fluorescent brightener 28 was used at a final concentration of 10 $\mu$g/ml to stain the fungal hyphae in preparation for confocal microscopy. To examine the effect of actin organization on the localization of chitin synthases, the actin inhibitor latrunculin A (Lat A) (Sigma-Aldrich) was used at a final concentration of 10 $\mu$g/ml to treat the various strains. Microscopic examinations were performed using a Nikon A1 laser confocal microscope (Nikon, Japan).

**Quantitative real-time PCR and whole-genome sequencing analysis.** For RNA extraction, the various strains were cultured in liquid CM at 28°C with constant shaking at 110 rpm for 2 days. Mycelial samples were then collected, and total RNA was extracted using the Eastep Universal RNA extraction kit (Promega, Shanghai, China). The PrimeScript RT reagent kit (TaKaRa) was used for cDNA synthesis; the transcription levels of the genes were detected using TB Green Premix *Ex Taq* II (TaKaRa) using specific primers (Table S2). $\beta$-Tubulin was used as an internal standard, and final data were calculated by the

cycle threshold ($2^{-\Delta\Delta CT}$) method (74). For whole-genome sequencing analysis, two independent spontaneous mutation strains of the Δ*Motps2* mutant were cultured in liquid CM at 28°C at 110 rpm for 3 days. Mycelial samples were collected, and total DNA samples from the spontaneous strains were extracted and then sent for further whole-genome sequencing analysis (Novogene, Beijing). The sequence reads were mapped onto the reference genome of strain 70-15 (75). SV (structure variance) and SNP/indel (single-nucleotide polymorphism/insertion-deletion) analyses were carried out to identify the mutation sites in the spontaneous mutation strains.

**Immunoblot assay.** For immunoblot assays, the mycelia of the strains involved were cultured in liquid CM at 28°C with constant shaking at 110 rpm for 2 days, after which total proteins were extracted from them as previously described (76). Samples were separated by 10% SDS-PAGE and further analyzed by Western blotting with anti-GFP antibody (1:5,000 GFP-tagged mouse monoclonal antibody [Mab]; Abmart).

**Yeast two-hybrid and BiFC assays.** Yeast two-hybrid assays were performed as previous described (77). Briefly, the cDNA sequences of *MoTPS1*, *MoTPS2*, and *MoTPS3* were amplified using the primers listed in Table S1, after which *MoTPS2* and *MoTPS3* were cloned into pGADT7-T, while *MoTPS1* and *MoTPS2* were cloned into pGBKT7-53 vectors. Coexpression of pGADT7-T and pGBKT7-53 was used as a positive control, while coexpression of pGADT7-T and pGBKT7-Lam served as the negative control. The BiFC assays were conducted as previously described (78). The coding regions and native promoters of *MoTPS1*, *MoTPS2*, and *MoTPS3* were amplified using the primers listed in Table S1. The fragments were cloned into pKNT-NYFP and pCX62-CYFP vectors and then cotransformed (MoTps2-NYFP+MoTps1-CYFP, MoTps3-NYFP+MoTps2-CYFP, MoTps3-NYFP+MoTps1-CYFP, MoTps3-NYFP+pCX62-CYFP, MoTps2-NYFP+pCX62-CYFP, pKNT-NYFP+MoTps2-CYFP, pKNT-NYFP+MoTps1-CYFP) into the wild-type 70-15 protoplasts. The positive transformants were screened by PCR, and confocal microscopic examinations were carried to identify the interactions of the TPS complex subunits.

**Quantification of trehalose and trehalose 6-phosphate.** For the measurement of trehalose and trehalose 6-phosphate, GC-MS analysis was performed as previously described with small modifications (16, 79). The strains were cultured in liquid CM at 28°C with shaking at 110 rpm for 3 days. Mycelia from the strain cultures were collected and ground into powder in liquid nitrogen. Thirty milligrams of each sample was separately introduced into 10-ml centrifuge tube, 800 $\mu$l methanol and 10 $\mu$l double-distilled water (ddH$_2$O) were added, and the mixture was incubated at 70°C for 20 min and finally centrifuged. The supernatants were collected, and 500 $\mu$l ddH$_2$O and 268 $\mu$l chloroform were added. The mixture was centrifuged at 12,000 rpm for 10 min, and the resulting supernatant was transferred into a 2-ml tube and dried in a freeze dryer. About 50 $\mu$l methoxyamine-pyridine mixture was added into the tube, and the mixture was kept at 30°C for 90 min. Afterwards, 80 $\mu$l BSTFA [*N*,*O*-Bis(trimethylsilyl)trifluoroacetamide] plus 1% TMCS (trimethylchlorosilane) was added, and the mixture was incubated at 65°C for 60 min. The final samples were transferred to a GC-MS glass vial and then analyzed with a Gerstel autosampler, an Agilent 7890B gas chromatograph, and Pegasus HT time-of-flight mass spectrometer (LECO Co., Saint Joseph, MI, USA). Trehalose 6-phosphate dipotassium salt (Santa Cruz) and D-(+)-trehalose dehydrate (Sigma-Aldrich) were used as standards.

**Statistical analysis.** Statistical analyses were performed using GraphPad Prism 7 (GraphPad Software, Inc., San Diego, CA). Data from three biological replicates were analyzed using the unpaired two-tailed Student's *t* test. Values are presented as means ± standard error (SE), where * indicates $P < 0.05$ and ** indicates $P < 0.01$.

## SUPPLEMENTAL MATERIAL

Supplemental material is available online only.

**FIG S1**, TIF file, 2 MB.
**FIG S2**, TIF file, 1.1 MB.
**FIG S3**, TIF file, 1.7 MB.
**FIG S4**, TIF file, 1.6 MB.
**FIG S5**, TIF file, 2.8 MB.
**FIG S6**, TIF file, 2.5 MB.
**FIG S7**, TIF file, 1.5 MB.
**TABLE S1**, DOCX file, 0.02 MB.
**TABLE S2**, DOCX file, 0.02 MB.
**TABLE S3**, DOCX file, 0.02 MB.

## ACKNOWLEDGMENTS

We thank Jin-Rong Xu (Purdue University, West Lafayette, IN, USA) for critical comments and Ying Xu (Zhejiang University, People's Republic of China) for transmission electron micrograph assays.

This research was supported by the National Natural Science Foundation of China (31772106) to W.Z. and the FAFU International Cooperation Project (KXB16010A) to J.Z.

The funders had no role in study design, data collection and analysis, decision to publish, or preparation of the manuscript.

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
