## [Reviewer comments · mSystems]

TPS complex-mediated regulation of trehalose 6-phosphate homeostasis is critical for development and pathogenesis in *Magnaporthe oryzae*

Xin Chen, Yakubu Abubakar, Chengdong Yang, Xiaxia Wang, Pengfei Miao, Mei Lin, Yuetong Wen, Qiuqiu Wu, Haoming Zhong, Yuping Fan, Meiru Zhang, Zonghua Wang, Jie Zhou, and Wenhui Zheng

Corresponding Author(s): Wenhui Zheng, State Key Laboratory for Ecological Pest Control of Fujian and Taiwan Crops

Review Timeline:

Submission Date:	April 14, 2021
Editorial Decision:	June 1, 2021
Revision Received:	August 30, 2021
Accepted:	September 8, 2021

Editor: Ileana Cristea

Reviewer(s): Disclosure of reviewer identity is with reference to reviewer comments included in decision letter(s). The following individuals involved in review of your submission have agreed to reveal their identity: Zhengguang Zhang (Reviewer #1); Rongyu Li (Reviewer #2)

Transaction Report:

DOI: <https://doi.org/10.1128/mSystems.00462-21>

June 1, 2021

Dr. Wenhui Zheng
State Key Laboratory for Ecological Pest Control of Fujian and Taiwan Crops
Fuzhou, Fujian 350002
China

Re: mSystems00462-21 (TPS complex-mediated regulation of trehalose 6-phosphate homeostasis is critical for development and pathogenesis in *Magnaporthe oryzae*)

Dear Dr. Wenhui Zheng:

Thank you for submitting your manuscript to mSystems. We have completed our review and I am pleased to inform you that, in principle, we expect to accept it for publication in mSystems. However, acceptance will not be final until you have adequately addressed the reviewer comments.

The reviewers suggested a number of additions and changes that will further strengthen the manuscript. In particular, one of the reviewers indicated that additional evidence may be necessary to demonstrate the T6P is induced during fungus development. We invite the submission of a revised manuscript that addresses the reviewers comments.

Preparing Revision Guidelines

For complete guidelines on revision requirements, please see the Instructions to Authors at <https://msystems.asm.org/sites/default/files/additional-assets/mSys-ITA.pdf>. **Submissions of a paper that does not conform to mSystems guidelines will delay acceptance of your manuscript.**

Corresponding authors may join or renew ASM membership to obtain discounts on publication fees. Need to upgrade your membership level? Please contact Customer Service at

Service@asmusa.org.

Sincerely,

Ileana Cristea

Editor, mSystems

Journals Department
Reviewer comments:

Reviewer #1 (Comments for the Author):

The manuscript describes the balance of T6P accumulation in the rice blast disease which caused the defects in the fungal invasion structure and virulence. The author found a self-mutation of its MoTPS3 caused the recovery on the deletion of MoTps2. The author design various methods to verify the hypothesis that the organisms maintain a safe T6P level and cope with its cytotoxicity effects by the self-mutation. The manuscript was well-written, however, in my point, the ms lack some key evidences for the conclusion which led to the misleading to the readers.

The most important tissue is when the T6P enhanced? In line 86-87, When trehalose is no more required (such as in the absence of 86 stress)? However, the author showed no evidence here that during the development of the fungus the T6P was induced. As this evidence missing, the author lacks the results that the Tps3 mutation in vivo. I do not agree to the conclusion in the MS that the self-mutation of TPs3 is required for the virulence.

Another major gap is that, how the tps3 mutation happened? NO evidence here. Are these sites active at the basic level even not in the deletion of MoTps2? The mutation of Tps3 is only dependent to Tps2 or once immature of the appressorium? After reading carefully throughout the Ms, I still can not follow that why the mutation occurred. I think the author should provided more evidence here.

Reviewer #2 (Comments for the Author):

Chen et al. investigated the mechanism by which the physiologic level of trehalose-6-phosphate (T6P) is regulated in the rice blast fungus *M. oryzae*. Despite excellent presentation, few

grammatical errors need to be fixed in the manuscript.

L128-132: You rather observed (not characterized) a spontaneous mutation that partially restores the defects. Also, this sentence is too long, making it difficult to grab the message. I suggest you break it into two simpler sentences.

L142: define aa (as it appears here for the very first time)

L149: '.....in the Motps2 mutant' please recast as '.....in the Motps2 mutant background'

L151: '.....to their compare their growth rates' please check and fix the error.

L236: '.....appropriate location of CHS genes is essential for cell wall homeostasis' (please provide a reference for this).

L272-273: under what condition(s) does the mutant produce spontaneous suppressor strain? This is necessary because I cannot see the strain emerging in other culture plates as in Fig. 1 A.

L595: Delete one 'Afterwards'.

The materials and methods section does not explain how the various forms of mutation that exist in the spontaneous strain were identified.

Response to Reviewer #1

The manuscript describes the balance of T6P accumulation in the rice blast disease which caused the defects in the fungal invasion structure and virulence. The author found a self-mutation of its MoTPS3 caused the recovery on the deletion of MoTps2. The author design various methods to verify the hypothesis that the organisms maintain a safe T6P level and cope with its cytotoxicity effects by the self-mutation. The manuscript was well-written, however, in my point, the ms lack some key evidences for the conclusion which led to the misleading to the readers.

Response

We thank the reviewer for the constructive criticism and time spent to analyze this manuscript. Our responses and explanations in relation to the comments are provided below. We hope that the changes we have made resolve all your concerns about the article. We are more than happy to make any further changes that will improve the paper and/or facilitate successful publication.

The most important tissue is when the T6P enhanced? In line 86-87, When trehalose is no more required (such as in the absence of 86 stress)? However, the author showed no evidence here that during the development of the fungus the T6P was induced. As this evidence missing, the author lacks the results that the Tps3 mutation in vivo. I do not agree to the conclusion in the MS that the self-mutation of TPs3 is required for the virulence.

Response

Thanks for your positive comments. Coupled with previous studies and our results, the accumulated T6P can only be found in the Δ *Motps2* mutant (~2000-fold) from the fungal mycelia. Both Δ *Motps1* and Δ *Motps3* mutants are impaired of T6P production. In Fig. 4 e and Fig.6 a, our results also indicated that the intracellular T6P maintains a relatively low level in the mycelium. The Δ *Motps2* mutant has severe defects in conidiation and appressorium formation. We could not obtain enough conidia and appressorial tissues for further GC-MS analysis. In *Fusarium graminearum*, loss of FgTps2 also leads to highly accumulated T6P level in both mycelia and conidia. It is possible that the conidia and appressoria of the Δ *Motps2* mutant also accumulated high level of T6P and thus leads to the observed severe defects.

MoTps1 is involved in the biosynthesis of T6P. To increase the intracellular level of T6P, we generated Δ *Motps1-OE* strains by overexpressing the trehalose 6-phosphate synthase *MoTPS1* in the wild type (Fig.R1 A). Although over-expression of *MoTPS1* leads to defects in growth, conidiation and virulence, the Δ *Motps1-OE* strain has no significant effect on T6P production (Fig.R1 B). Since trehalose biosynthesis is induced when exposed to environmental stresses, we further monitored the expression level of the TPS complex subunits through qRT-PCR assays. We noticed that the expression levels of *MoTPS1* and *MoTPS3* increased by ~1.8-fold under oxidative stress (Fig.R1 C). In the Δ *Motps1-OE* strains, the expression level of the *MoTPS1* was

up-regulated by ~10-fold, while there was only little increase in the T6P level. These results also pushed us to investigate how *M. oryzae* physiologically overcomes the stress due to T6P accumulation. Interestingly, we noticed that the $\Delta Motps2$ mutant undergoes spontaneous mutation in its *MoTPS3* gene to take care of the accumulated T6P. Further results indicated that the spontaneous mutants have low level of intracellular T6P. This finding indicates an important role of the MoTps3 in modulating T6P level. Based on these results, we suggested that *M. oryzae* maintains a low level of T6P under normal condition, and the T6P level could increase under stress condition.

Fig.R1 Over-expression of *MoTPS1* has no significant effect on the production of trehalose and trehalose 6-phosphate

(A) Over-expression of *MoTPS1* leads to defects in growth, conidiation and virulence. (B) The $\Delta Motps1$ -OE strain has no significant effects on trehalose and T6P production. (C) Expression levels of the TPS complex subunits under treatment with H₂O₂.

Trehalose is a non-reducing disaccharide that protects proteins and cellular membranes from inactivation or denaturation caused by a variety of stress conditions. In response to specific environmental stresses, activities of the TPS complex subunits are increased and thus induce trehalose biosynthesis. With the recovery of stress, trehalose biosynthesis is no more required and trehalose needs to be degraded. Degradation of trehalose by trehalases provides the energy necessary for stress recovery. We are sorry that the sentence in L86-87 makes you confused and this sentence has been revised as “Under stress recovery condition, trehalose biosynthesis is no more required” in the revised manuscript.

In 2007, Wilson RA *et.al* indicated that $\Delta Motps3$ mutant is unable to cause rice blast disease

(Wilson, Jenkinson et al. 2007), this can be found in the introduction part in L101-103. Based on this finding, we did not bother to generate $\Delta Motps3$ mutant. Since our study mainly focuses on the effects of mutation in *MoTPS3* on T6P level, we generated the $\Delta Motps3\text{-}\Delta Motps2$ double deletion mutant, and further assays indicated that deletion of *MoTPS3* in the $\Delta Motps2$ mutant background partially restores $\Delta Motps2$ mutant defects in growth, conidiation and virulence. In addition, the disease lesions caused by the $\Delta Motps3\text{-}\Delta Motps2$ double deletion mutant were significantly smaller than those caused by the wild type (Fig.4, Table 2). Based on these results, we concluded that self-mutation in *MoTPS3* down-lowers T6P level and thus partially rescues $\Delta Motps2$ mutant defects.

(D) Guy11, $\Delta tps1$ and $\Delta tps3$ strains were inoculated onto rice seedlings to analyse pathogenicity. Spores were inoculated at a rate of 5×10^4 spores/ml

Wilson, R. A., J. M. Jenkinson, R. P. Gibson, J. A. Littlechild, Z. Y. Wang and N. J. Talbot (2007). "Tps1 regulates the pentose phosphate pathway, nitrogen metabolism and fungal virulence." EMBO J 26(15): 3673-3685.

Another major gap is that, how the *tps3* mutation happened? NO evidence here.

Response

This is actually the very first work that unveils the occurrence of this mutation due to persistent T6P accumulation in *M. oryzae*. As such, we decided to dissect in details how this spontaneous mutation occurs and the mechanism by which T6P induces the mutation in our future studies. However, we discussed some hypotheses that could possibly help to uncover this mechanism to aid further work. Spontaneous mutation is ultimate source of genetic variation and a fundamental component of evolution. Spontaneous mutation rate tends to rise when they are under antibiotic treatment, starvation, or other stresses. These genetic variations in turn provide beneficial

evolutionary changes for adaption (Flynn, Chain et al. 2017, Liu and Zhang 2019, Ho, Macrae et al. 2020). Genes encoding the proteins involved in trehalose biosynthesis are mechanistically linked to metabolism, cell wall homeostasis, stress responses, and virulence (Thammahong, Puttikamonkul et al. 2017). Our results indicated that the Δ *Motps2* mutant displayed high sensitivity to cell wall stress (Fig.3) and rapamycin (unpublished data). Moreover, the accumulated T6P in Δ *Motps2* mutant disrupt metabolite homeostasis (Fig.6). These results suggested that the accumulated T6P in Δ *Motps2* mutant disrupts cellular homeostasis and could serve as a stress agent that induces the genetic variation. In *M. oryzae*, MoTps1 is responsible for the biosynthesis of T6P, and MoTps3 physically interacts with MoTps1 (Fig.5b). It is possible that the accumulated T6P in Δ *Motps2* mutant can be sensed by MoTps1-MoTps3 and therefore induces the mutation in MoTps3.

In 2014, Song *et al.* indicated in *Fusarium graminearum* that the trehalose 6-phosphate phosphatase FgTps2 is required for fungal development, virulence and mycotoxin production (Song, Li et al. 2014). Loss of FgTps2 also leads to severe intracellular accumulation of T6P. To better understand how the accumulated T6P induces spontaneous mutation (and to satisfactorily address the reviewer's concern), we generated Δ *Fgtps2* mutant in *F. graminearum*. As shown in Fig. R2, the Δ *Fgtps2* mutant also produced spontaneous mutation strains when cultured on SYM medium, similar to what we demonstrated in *M. oryzae*. A previous study suggested in *F. graminearum* that the observed defects of Δ *Fgtps1*- Δ *Fgtps2* mutant were less severe when compared to those in Δ *Fgtps2* mutant (Song, Li et al. 2014) (Table R1). In addition, both trehalose and T6P production were abolished in the Δ *Fgtps1*- Δ *Fgtps2* mutant. These results indicated that the accumulated T6P could also act as a spontaneous mutation inducer in response to intracellular T6P accumulation and *FgTPS1* or *FgTPS3* could act as the potential mutation targets in *F. graminearum*.

Fig.R2 Spontaneous mutation of Δ *Fgtps2* mutant in *Fusarium graminearum*. The fast-growing sectors were marked by red arrows.

Table 1Conidiation, spore character, chitin content, chitin synthase activity and hyphal cell wall of the wild-type strain 5035, three null mutants and two complementation strains^a.

Strain	Conidiation (10 ⁵ ml ⁻¹) ^b	Conidial spores (μm) ^c			No. of septa ^d	Hyphal width (μm) ^e	Chitin content (%) ^f	Chitin synthase activity (nmol h ⁻¹ mg ⁻¹) ^g	Cell wall width (nm) ^h
		Length	Width	Length/width					
5035	9.83 ± 1.40a	40.31 ± 0.50a	4.37 ± 0.06a	9.41 ± 0.15a	3.66 ± 0.14a	5.00 ± 0.10b	25.02 ± 0.50a	0.45 ± 0.02a	135.05 ± 18.79b
Δtps1	8.02 ± 1.45a	38.68 ± 0.47a	4.29 ± 0.04a	9.06 ± 0.11a	3.34 ± 0.11a	5.27 ± 0.03b	24.34 ± 1.52a	0.44 ± 0.02	120.36 ± 18.92b
TPS1C	9.79 ± 1.6a	38.16 ± 0.71a	4.18 ± 0.06a	9.12 ± 0.12a	3.37 ± 0.12a	5.37 ± 0.14b	22.84 ± 0.57a	0.47 ± 0.05a	147.27 ± 50.3b
Δtps2	0	29.46 ± 0.51b	3.98 ± 0.05b	7.49 ± 0.14b	0.82 ± 0.17b	6.92 ± 0.30a	18.61 ± 0.64b	0.31 ± 0.03b	380.51 ± 52.21a
Δtps2 [*]	0.93 ± 0.11b								
TPS2C	10.53 ± 0.42a	39.92 ± 0.43a	4.21 ± 0.04a	9.56 ± 0.12a	3.40 ± 0.13a	5.20 ± 0.09a	22.83 ± 0.69a	0.49 ± 0.02a	146.30 ± 31.55b
Δtps1-Δtps2	8.00 ± 0.94a	40.59 ± 0.51a	4.29 ± 0.06a	9.64 ± 0.14a	3.37 ± 0.10a	5.10 ± 0.13a	23.46 ± 0.64a	0.48 ± 0.05a	125.42 ± 22.37b

^a Data in this table are represented as averages ± standard errors. Different letters represent a significant difference at $P < 0.01$.^b Macroconidia were counted after being cultured in CMC or CMC + 20% PEG4000 (*) for 5 d. Experiments were performed in triplicate.^c The length and width of 200 spores randomly selected from each strain were measured.^d Septa were counted in 200 spores staining with CFW (10 μg ml⁻¹).^e Strains were cultured in CMC with 20% PEG4000 for 3 d at 28 °C (200 rpm) under constant fluorescent lighting. Hyphal width was counted using 100 hyphae in three replicates.^f Chitin content was measured 24 h post-inoculation with macroconidia and was expressed as a percentage of N-acetylglucosamine (GlcNAc) relative to dry mycelial weight. The experiment was performed in triplicate.^g Chitin synthase activity was measured 48 h post-inoculation with macroconidia. Data are expressed as nanomoles of GlcNAc incorporated per hour per milligram of protein from three replicates.^h Hyphae were observed and photographed with transmission electron microscope. The cell wall width of 60 cells randomly selected from each strain was measured.

Table R1 Conidiation, spore character, chitin content, chitin synthase activity and hyphal cell wall of the wild-type strain 5035, three null mutants and two complementation strains (Song, Li *et al.* 2014)

Flynn, J. M., F. J. Chain, D. J. Schoen and M. E. Cristescu (2017). "Spontaneous Mutation Accumulation in *Daphnia pulex* in Selection-Free vs. Competitive Environments." *Mol Biol Evol* 34(1): 160-173.

Ho, E. K. H., F. Macrae, L. C. Latta, P. McIlroy, D. Ebert, P. D. Fields, M. J. Benner and S. Schaack (2020). "High and Highly Variable Spontaneous Mutation Rates in *Daphnia*." *Mol Biol Evol* 37(11): 3258-3266.

Liu, H. and J. Zhang (2019). "Yeast Spontaneous Mutation Rate and Spectrum Vary with Environment." *Curr Biol* 29(10): 1584-1591 e1583.

Song, X. S., H. P. Li, J. B. Zhang, B. Song, T. Huang, X. M. Du, A. D. Gong, Y. K. Liu, Y. N. Feng, R. S. Agboola and Y. C. Liao (2014). "Trehalose 6-phosphate phosphatase is required for development, virulence and mycotoxin biosynthesis apart from trehalose biosynthesis in *Fusarium graminearum*." *Fungal Genet Biol* 63: 24-41.

Thammahong, A., S. Puttikamonkul, J. R. Perfect, R. G. Brennan and R. A. Cramer (2017). "Central Role of the Trehalose Biosynthesis Pathway in the Pathogenesis of Human Fungal Infections: Opportunities and Challenges for Therapeutic Development." *Microbiol Mol Biol Rev* 81(2): e00053-16(2).

Are these sites active at the basic level even not in the deletion of *MoTps2*? The mutation of *Tps3* is only dependent to *Tps2* or once immature of the appressorium? After reading carefully throughout the Ms, I still cannot follow that why the mutation occurred. I think the author should provide more evidence here.

Response

As shown in Fig.R3, the spontaneous mutation sites identified in *MoTPS3* are well conserved in different species. Our results also indicated that the spontaneous mutation strains $\Delta MoTps2$ -*m* mutant has similar phenotypes when compared to the $\Delta MoTps2$ - $\Delta MoTps3$ mutant (Fig.4). It is possible that these mutations sites could be induced under other stress conditions. This is also a topic of further studies. As for the existing evidences, the spontaneous mutation is observed when

MoTPS2 gene is deleted and this partially ameliorates the defects of the *tps2* mutant. This therefore simply suggests that using MoTps2 as a drug target to combat the rice blast infection could be compromised by this spontaneous mutation of *MoTPS3*. Whether there are other conditions that can lead to this mutation apart from the deletion of *MoTPS2* will make part of an interesting subject of our next research as this will shade more light on the feasibility of targeting the TPS complex for antifungal drug discovery.

Tps3p	MTIIIVASLFLYPTQFEADVNSDTAKLVESMIKVCDCNQCENLSSNNKQERSSSSVISASSHYIGLPQEAQINGEPLQRANVGPATGVNYHNEMEMLSSEQ	100
MoTps3	0
NcTps3	0
FgTps3	0
VdTps3	0
Consensus		
Tps3p	FLEELTANATHANGSIFANNFVSSGTAQRFSVEEFFSAPSAVRCVSPSQEASASSISA SRSSAHNHLSSSLMKNPNLNSFDSHFPRVSRSSKSAVITP	200
MoTps3MTVEIASLFLFKTVHHERLFCGFS.....RSPFPALDKDRAMGDVPAAGGLK.....SFSHAKQPS.....	55
NcTps3MTVEIASLFLFKTVHHERLFCGFS.....RGSASQKFAKAMKLAEN.....TFSLFCP.....	50
FgTps3MTVEVCSLFLFKTIHHTLFCGTPPPGVDSRSTLSSGSSNNLNRGDTASLSGSAVKKGDAA.....LPPKTAAPVGGRAAPLNRCPS....LF	83
VdTps3MTVEVATLFLFKTVHHERLFCGFS.....GTAIPAPERSERTIKKPGESTLTG.....ADKSLFTPLP.....	56
Consensus	p g p	
Tps3p	VSKSVFDVDFVVDVAKVREEFCCQASLPSMKRVSGST.AGDSSTASSSSNLRYSQQCFDNFIEDTSDSEDDIDSDLETDATKKYNVFKCGYSNNAKLRA	299
MoTps3	LFRDITPPETHIEERPAEADIFANEDGLAVQFPAPTD.AGKQGFADRGSEPTWAGARRDQFMSRANSFPPF..SLLNHNKTLALKAREIGRMGVKQFLSL	152
NcTps3	AFSHITPPHTTEEDR.KHHPWANEDGLKVCIFQMPFFSPDGSYRAFIDRSSPTWGGRRFDQFMSRANSFPPF..FSLINSRRALNCKAREICROGITQFRSL	147
FgTps3	QKEDITPPRTPEEV.DVN.LFANEDGIRIPFFKR.....FGSNAGSGKE.WGSRANQPKSRASFPFPF..PALSENSRTM.QKAREIGRQGVVQKFKFL	170
VdTps3	...EVTPEDTTEEH.GSQDLFANEDGFRIFLSDDGNE.SVDAGMPADKGRSEWGGSANQPKSRASFPFPF..ALIAEHSRTL.EKAREIGRQGVVQKFRAT	148
Consensus	g	
Tps3p	SLMRNSYELFKHLPTWVDSFKGNSLNNVNIWAAEKTVKEPVSWYGTGCTPDPF.P.HEVCHKISKPECDFFSSFFVTDITFKGAKNYAKKILWP	398
MoTps3	TRSDSHDRVFAQADYTIWVPCGNGGLNNAEPAAREGLGD.HTYVGTGCTPDPFDGTQCKQDIEDRLATEHDCIAPWCSKDFGEGYSHFGKQKILWP	251
NcTps3	VRSESHDRVFAHADKVIWVSDGCGNGGLNNAEPAARDGKLG.E.YTWVGTGCTPDPFRTGQQLQDIDDLRATERHDLAVFCSDKDFGEGYSHFGKQKILWP	246
FgTps3	VRSDSHDRVFAASAGMVVNDGCGNGGLNNAEPAARDGRIDE.ITWVGTGCTPDPFEGTEQCKQDIENTLANEHNMITVFCSDKDFGEGYAHFGKQKILWP	269
VdTps3	ARSDSHDRVFAHANKVWVNDGCGNGGLNNAEPAARDGKLG.E.YTWVGTGCTPDPFEGTEQCKQDIEDHLATDHDMLITVFCSDKDFGEGYSHFGKQKILWP	247
Consensus	f v d gng l na a gt g ptd l i l v d f g y k ilwp	
Tps3p	LIHYQIPDNFSKAEEDHSWYVYVNCVAFADIVKNNKRGVWVHDYHLLVVEGMRKRFPAKIGFFLHVFPSSSEVFRCLAVRRELLEGMLGANLV	498
MoTps3	VHYQIPDNFSKAEEDHSWYVYVNCVAFADIVKNNKRGVWVHDYHLLVVEGMRKRFPAKIGFFLHVFPSSSEVFRCLAVRRELLEGMLGANLV	351
NcTps3	VHYQIPDNFSKAEEDHSWYVYVNCVAFADIVKNNKRGVWVHDYHLLVVEGMRKRFPAKIGFFLHVFPSSSEVFRCLAVRRELLEGMLGANLV	346
FgTps3	VHYQIPDNFSKAEEDHSWYVYVNCVAFADIVKNNKRGVWVHDYHLLVVEGMRKRFPAKIGFFLHVFPSSSEVFRCLAVRRELLEGMLGANLV	369
VdTps3	VHYQIPDNFSKAEEDHSWYVYVNCVAFADIVKNNKRGVWVHDYHLLVVEGMRKRFPAKIGFFLHVFPSSSEVFRCLAVRRELLEGMLGANLV	347
Consensus	hyqipdnp ska edhsw yy v nq d i k gd w hdyhl lvp m r k p akigfflhv fpsssevfrcla r leg gan	
Tps3p	GFCEIYGRHRLQTCRRLDAEATDGLQLEDREVDVNNLFGIDPVSLSRHRGSEVWRWLDIMRERYACRRLIVARKEKIDHVRGVRCKLSYEFFLNK	598
MoTps3	GFCEIYGRHRLQTCRRLDAEATDGLQLEDREVDVNNLFGIDPVSLSRHRGSEVWRWLDIMRERYACRRLIVARKEKIDHVRGVRCKLSYEFFLNK	451
NcTps3	GFCEIYGRHRLQTCRRLDAEATDGLQLEDREVDVNNLFGIDPVSLSRHRGSEVWRWLDIMRERYACRRLIVARKEKIDHVRGVRCKLSYEFFLNK	446
FgTps3	GFCEIYGRHRLQTCRRLDAEATDGLQLEDREVDVNNLFGIDPVSLSRHRGSEVWRWLDIMRERYACRRLIVARKEKIDHVRGVRCKLSYEFFLNK	469
VdTps3	GFCEIYGRHRLQTCRRLDAEATDGLQLEDREVDVNNLFGIDPVSLSRHRGSEVWRWLDIMRERYACRRLIVARKEKIDHVRGVRCKLSYEFFLNK	447
Consensus	gfgq ey rhf qcc r l d v v igid v w k liv rd d rg k l ye fl	
Tps3p	NPEYIEKVVLLICICIGKSDPEYERCMVVDVIRNSLSSNISIQVWVHFDLDFACYLALNCEAEVFLVDAIREGMNLTGHEFVSS.....FERNA	692
MoTps3	NPEWRGKTMILICVALSTSEKSELDAVSDVIRVNSWASLTY.QPVTMLRQDIDGYCYLALTIADALMITSQREGMNLTSHEFLACDQKNSNDRKRG	550
NcTps3	NPEWRGKTMILICVALSTSEKSELDAVSDVIRVNSWANLAY.QPVTMLRQDIDGYCYLALTIADALMITSQREGMNLTSHEFLACDQKNSNDRKRG	544
FgTps3	NPEWRNTVLLICVALSSSEKSDLEATVSDVIRVNSWANLAY.QPVTMLRQDIDGYCYLALTIADALMITSQREGMNLTSHEFLACDQKNSNDRKRG	568
VdTps3	NPEWRNCTVLLICVAMSSSEKSDLEAASVSDVIRVNSWANLAY.QPVTMLRQDIDDFCYLALTIADALMITSQREGMNLTSHEFLACDQKNSNDRKRG	546
Consensus	np l i q s v r ns qv l qd qylal ad regmnlts he k	
Tps3p	HLLSEFTGSSVLEKGAAILNFWINHWACSTKRSLEMSPEERARRNKLEKSVIEHSDVWITKCFEYINNAWESNG..ETSTFNFAPEKFCADVYA	790
MoTps3	SLLSEFTGSSVLEKGAAILNFWINHWACSTKRSLEMSPEERARRNKLEKSVIEHSDVWITKCFEYINNAWESNG..ETSTFNFAPEKFCADVYA	650
NcTps3	SLLSEFTGSSVLEKGAAILNFWINHWACSTKRSLEMSPEERARRNKLEKSVIEHSDVWITKCFEYINNAWESNG..ETSTFNFAPEKFCADVYA	644
FgTps3	SLLSEFTGSSVLEKGAAILNFWINHWACSTKRSLEMSPEERARRNKLEKSVIEHSDVWITKCFEYINNAWESNG..ETSTFNFAPEKFCADVYA	668
VdTps3	SLLSEFTGSSVLEKGAAILNFWINHWACSTKRSLEMSPEERARRNKLEKSVIEHSDVWITKCFEYINNAWESNG..ETSTFNFAPEKFCADVYA	646
Consensus	l lseftg ss v l e k g a i l n f w i n h w a c s t k r s l e m s p e e r a r r n k l e k s v i e h s d v w i t k c f e y i n n a w e s n g . . e t s t f n f a p e k f c a d v y a	
Tps3p	SKKHLFFFKIS.....EPPTSMLSLFSSSN..NLYVVSFTRKNTFESLYN.GVNIIGLIAEAGAYVRVNG..SKYNIIVEE...LDMK	869
MoTps3	SSKRLFLDYEGTIVSWGPNQIIFVSPQRTLLVNSDILLDEQNTVYVSGRRPEELDRFRARIRNIGLIAEAGCGLLRKCGADSAEMADAGHNRNHE	750
NcTps3	TNRRLLFDDEFTLVSWGPNQIIFVSPQRTLLVNSDILLDEQNTVYVSGRRPEELDRFRFRVNIIGLIAEAGCGLLRKCGADSAEMADAGHNRNHE	743
FgTps3	AERRLFDYEGTIVSWGPNQIIFVSPQRTLLVNSDILLDEQNTVYVSGRRPEELDRFRFRVNIIGLIAEAGCGLLRKCGADSAEMADAGHNRNHE	767
VdTps3	SERRLLFDYEGTIVSWGPNQIIFVSPQRTLLVNSDILLDEQNTVYVSGRRPEELDRFRFRVNIIGLIAEAGCGLLRKCGADSAEMADAGHNRNHE	745
Consensus	lfi ldyegtivswgpnqiifvspqrtl llv n s d i l l d e q n t v y v s g r r p e e l d r f r v n i i g l i a e a g c g l l r k c g a d s a e m a d a g h n r n h e	
Tps3p	EVAKIFDEKVFEPFSYKIDSMIRHTENRFDQRVFVIGEAITHNLFDRDRDHHVYHKDIVFQQTGLAL.AAEFLMKFYNSGVSPIDNSRIS	968
MoTps3	SVRGMITYFERPFGAEVEERRCSLVFHYKIDSMIRHTENRFDQRVFVIGEAITHNLFDRDRDHHVYHKDIVFQQTGLAL.AAEFLMKFYNSGVSPIDNSRIS	843
NcTps3	SLRDIMTYFERPFGAEVEERRCSLVFHYKIDSMIRHTENRFDQRVFVIGEAITHNLFDRDRDHHVYHKDIVFQQTGLAL.AAEFLMKFYNSGVSPIDNSRIS	843
FgTps3	SVRGMITYFERPFGAEVEERRCSLVFHYKIDSMIRHTENRFDQRVFVIGEAITHNLFDRDRDHHVYHKDIVFQQTGLAL.AAEFLMKFYNSGVSPIDNSRIS	867
VdTps3	SVRSIMTYFERPFGAEVEERRCSLVFHYKIDSMIRHTENRFDQRVFVIGEAITHNLFDRDRDHHVYHKDIVFQQTGLAL.AAEFLMKFYNSGVSPIDNSRIS	845
Consensus	i er pg fh a d h n ha v aa	
Tps3p	LSRTSSMSVGNKKHFQNVDFVCSGTSSTFIEPLKLVQVEKRNKIKFYITIIYSSRSRYAKEHINGVNELFTILHDLTAA.	1054
MoTps3	FD.FLMVVGGRDEKVEKWANLIGEBCTVKDVVIVSLGSR.NTEARATITGCVSGVITAIQKLAALS.....	909
NcTps3	VD.FLMVVGGRDEKVEKWANLIGEBCTVKDVVIVSLGSR.NTEARATITGCVSGVITAIQKLAALS.....	928
FgTps3	VD.FLMVVGGRDEKVEKWANLIGEBCTVKDVVIVSLGSR.NTEARATITGCVSGVITAIQKLAALS.....	933
VdTps3	VD.FLMVVGGRDEKVEKWANLIGEBCTVKDVVIVSLGSR.NTEARATITGCVSGVITAIQKLAALS.....	911
Consensus	g g g	

Fig.R3 Amino acids alignment of Tps3 homologues in different species. The base deletion mutation site was marked by red cycle and the base insertion site was marked by blue arrow.

Response to Reviewer #2

We would like to thank the editorial board and the reviewers for their constructive comments concerning this article (control no. mSystems00462-21 R0). These comments are all valuable and helpful for improving our article. All the authors have seriously discussed about all these comments. We therefore modified the manuscript accordingly to accommodate the reviewer's observations and meet the requirements of the journal. Point-by-point responses to the reviewer's comments are listed below.

L128-132: You rather observed (not characterized) a spontaneous mutation that partially restores the defects. Also, this sentence is too long, making it difficult to grab the message. I suggest you break it into two simpler sentences.

Response

Thanks a lot for your valuable suggestion to improve the quality of our manuscript. The sentence has been revised as “In addition, the TPS complex possesses a spontaneous ‘correction’ way to modulate T6P homeostasis by spontaneous mutation of the regulatory subunit MoTps3. This spontaneous correction function further suppresses the MoTps1 activity to down-regulate T6P, resulting in partial restoration of the Δ *Motps2* mutant defects in growth, conidiation and pathogenicity.”

L142: define aa (as it appears here for the very first time)

Response

Thanks for your careful checks. We have defined “aa” in L142 in the revised manuscript.

L149: '.....in the Motps2 mutant' please recast as '.....in the Motps2 mutant background'

Response

Thanks for your suggestion, the “in the Δ *Motps2* mutant” has been changed to “in the Δ *Motps2* mutant background” in L149.

L151: '.....to their compare their growth rates' please check and fix the error.

Response

We are sorry for this grammatical error. The sentence has been revised as “The wild type (70-15), mutant (Δ *Motps2*) and complemented strain (Δ *Motps2-com*) were then cultured on CM, MM and RBM culture media for 7 days to compare their growth rates.” The revision is shown in L149-151.

L236: '.....appropriate location of CHS genes is essential for cell wall homeostasis' (please provide a reference for this).

Response

Thanks for this important comment; we have inserted the related references in L236. The sentence was revised as “Since the appropriate localization of CHS genes is essential for cell wall homeostasis (29, 39)”. The references are listed as follow:

29. Riquelme M. 2013. Tip growth in filamentous fungi: a road trip to the apex. *Annu Rev Microbiol* 67:587-609.

39. Thammahong A, Caffrey-Card AK, Dhingra S, Obar JJ, Cramer RA. 2017. *Aspergillus fumigatus* Trehalose-Regulatory Subunit Homolog Moonlights To Mediate Cell Wall Homeostasis through Modulation of Chitin Synthase Activity. *mBio* 8: e00056-17.

L272-273: under what condition(s) does the mutant produce spontaneous suppressor strain? This is necessary because I cannot see the strain emerging in other culture plates as in Fig. 1 A.

Response

As shown in the Fig.S7, the Δ *Motps2* mutant produced spontaneous suppressor strains when we revived Δ *Motps2* mutant from filter papers on SYM medium. Based on our experience, the Δ *Motps2* mutant produced spontaneous suppressor strains randomly. This happened when we performed the growth assays, cell wall stress assays, osmotic assays, etc. In our manuscript, the first part of our results section is mainly about the biological function of the MoTps2 (Fig.1, Fig.2 and Fig.3), while Fig.4, Fig.5 and Fig.6 focus on the identification of the mutation sites in the spontaneous suppressor strain and how the mutations of *MoTPS3* restore the Δ *Motps2* mutant's defects.

L595: Delete one 'Afterwards'.

Response

Thanks for your correction; we have deleted one “afterwards” in L595. We are sorry for this mistake.

The materials and methods section does not explain how the various forms of mutation that exist in the spontaneous strain were identified.

Response

Thanks for your valuable suggestion. How various forms of mutation in the spontaneous strain were identified was added in the materials and methods part. This can be seen in L553-560 in the revised manuscript.

September 8, 2021

Dr. Wenhui Zheng
State Key Laboratory for Ecological Pest Control of Fujian and Taiwan Crops
Fuzhou, Fujian 350002
China

Re: mSystems00462-21R1 (TPS complex-mediated regulation of trehalose 6-phosphate homeostasis is critical for development and pathogenesis in *Magnaporthe oryzae*)

Dear Dr. Wenhui Zheng:

Congratulations! Your manuscript has been accepted for publication in mSystems. Thank you for your careful consideration of the reviewers' comments.

As your manuscript has been accepted, I am forwarding it to the ASM Journals Department for publication. For your reference, ASM Journals' address is given below. Before it can be scheduled for publication, your manuscript will be checked by the mSystems senior production editor, Ellie Ghatineh, to make sure that all elements meet the technical requirements for publication. She will contact you if anything needs to be revised before copyediting and production can begin. Otherwise, you will be notified when your proofs are ready to be viewed.

As an open-access publication, mSystems receives no financial support from paid subscriptions and depends on authors' prompt payment of publication fees as soon as their articles are accepted. =

Publication Fees:

We recognize that the video files can become quite large, and so to avoid quality loss ASM suggests sending the video file via <https://www.wetransfer.com/>. When you have a final version of the video and the still ready to share, please send it to Ellie Ghatineh at eghatineh@asmusa.org.

Sincerely,

Ileana Cristea
Editor, mSystems

Journals Department
Fig. S3: Accept
Fig. S2: Accept
Fig.S7: Accept
Fig. S5: Accept
Fig. S1: Accept
Table S2: Accept
Fig. S4: Accept
Table S1: Accept
Fig. s6: Accept
Table S3: Accept